# Unravelling the neurocognitive mechanisms underlying counterconditioning in humans

Lisa Wirz[1,2]*[†], Maxime C Houtekamer[1†], Jette de Vos[3], Joseph E Dunsmoor[4], Judith Homberg[1‡], Marloes JAG Henckens[1‡], Erno Hermans[1]

[1]Donders Institute for Brain, Cognition, and Behaviour, Radboud University Medical Centre, Nijmegen, Netherlands; [2]Cognitive Psychology, Ruhr University Bochum, Bochum, Germany; [3]Department of Psychiatry and Neuropsychology, MHeNs, Maastricht University, Maastricht, Netherlands; [4]Department of Psychiatry and Behavioral Sciences, University of Texas at Austin, Austin, United States

**\*For correspondence:**
lisa.wirz@donders.ru.nl

[†]These authors contributed equally to this work
[‡]These authors also contributed equally to this work

**Competing interest:** The authors declare that no competing interests exist.

## eLife Assessment

This **important** work combines self-report, neural and physiology data to examine the efficacy and mechanisms of counter conditioning versus extinction in reducing re-emergence of conditioned threat responses and show that this appears to rely on the nucleus accumbens rather than the ventromedial prefrontal cortex. These findings are supported by **convincing** evidence, though some areas could benefit from a few targeted refinements. The findings will be of interest to researchers across multiple subfields, including neuroscientists, cognitive theory researchers, and clinicians, particularly those with an interest in clinical applications in trauma therapies.

**Abstract** Counterconditioning (CC) aims to enhance extinction of threat memories by establishing new associations of opposite valence. While its underlying neurocognitive mechanisms remain largely unexplored, previous studies suggest qualitatively different mechanisms from regular extinction. In this functional MRI study, participants underwent categorical threat conditioning (CS+/CS-: images of animals/tools), followed by either CC (CS + images reinforced with monetary rewards, n=24) or regular extinction (n=24). The following day, we assessed spontaneous recovery of threat responses and episodic memory for CS + and CS- category exemplars. While the ventromedial prefrontal cortex (vmPFC) was activated during regular extinction, participants undergoing CC showed persistent CS+-specific deactivation of the vmPFC and hippocampus, and CS+-specific activation of the nucleus accumbens (NAcc). The following day, physiological threat responses returned in the regular extinction group, but not in the CC group. Counterconditioning furthermore strengthened episodic memory for CS + exemplars presented during CC, and retroactively also for CS + exemplars presented during the threat conditioning phase. Our findings confirm that CC leads to more persistent extinction of threat memories, as well as altered consolidation of the threat conditioning episode. Crucially, we show a qualitatively different activation pattern during CC versus regular extinction, with a shift away from the vmPFC and towards the NAcc.

## Introduction

Trauma-related disorders are prevalent and highly detrimental to the individual's quality of life (*Kessler et al., 2005*). To treat these disorders, patients undergo exposure therapy in a safe therapeutic environment, causing threat responses to fade away (*Scheveneels et al., 2016*). Although

exposure therapy may be successful initially, relapse often occurs and is the most prevalent remaining challenge in optimizing treatment efficacy. Research suggests that exposure therapy creates a safety memory that competes for expression with the original threat memory (*Bouton, 2004*, *Myers and Davis, 2002*), suggesting that relapse may occur because of relatively weak learning and retention of the safety memory. Therefore, identifying mechanisms that can be used to strengthen safety learning is a key step in advancing treatment for trauma-related disorders. A promising approach to strengthen safety learning is to create a new, positive association with the event that was previously linked to an aversive outcome. However, while there are indications that establishing positive associations can prevent relapse, the underlying mechanisms are poorly understood (for a review, see *Keller et al., 2020*).

To study threat responses in a controlled setting, aversive Pavlovian conditioning is typically used. A neutral stimulus (conditioned stimulus, CS; e.g. a picture) is coupled with a biologically aversive unconditioned stimulus (US; e.g. an electrical shock), after which the CS alone also elicits a conditioned threat response. Conditioned threat responses to the CS can be attenuated using extinction, during which the CS is repeatedly presented in absence of the US. However, early theories have suggested that threat responses may more easily be inhibited by engaging appetitive systems (*Dickinson and Pearce, 1977*; *Rescorla and Solomon, 1967*). Indeed, experiments provide evidence that coupling a CS to a positive US after threat conditioning, a process known as aversive-to-appetitive CC, may be superior to regular extinction. Specifically, CC compared to regular extinction was associated with a faster attenuation of learned threat responses (*Dickinson and Pearce, 1977*; *Pearce and Dickinson, 1975*), stronger decreases in threat expectancy (*Kang et al., 2018*, *Newall et al., 2017*), and more positive valence ratings of the CS (*Luck and Lipp, 2018*; *van Dis et al., 2019*; *Jozefowiez et al., 2020*) immediately post-CC.

Tests for spontaneous recovery, reinstatement, and renewal can subsequently be used to evaluate the return of threat responses over time, after unsignaled presentation of the US, or in a novel context, respectively (*Bouton, 2004*, *Bouton, 2002*). Thereby, one can investigate whether CC persistently attenuates threat responses. While early rodent studies showed that CC may be prone to the same relapse as extinction (*Bouton and Peck, 1992*, *Brooks et al., 1995*), recent neurobiological work in rodents showed that CC can enhance the activation of an amygdala-striatal pathway, which is also recruited during extinction – albeit to a lesser degree – and that CC compared to regular extinction can reduce the return of threat responses (*Correia et al., 2016*). Recent studies suggest that CC may diminish the return of threat responses in humans as well. Specifically, it was shown that CC, compared to regular extinction, reduced renewal of previously learned food-allergy associations when presented in a novel context one day later (*Keller et al., 2023*). Counterconditioning compared to regular extinction was also associated with reduced recovery of arousal and shock expectancy the following day (*Kang et al., 2018*, *Keller et al., 2022*), as well as reduced reinstatement (*Kang et al., 2018*).

Extinction learning appears to be mediated by activation of the ventromedial prefrontal cortex (vmPFC), which inhibits the expression of threat responses by suppressing amygdala activity (*Quirk et al., 2000*; *Morgan et al., 1993*; *Quirk et al., 2003*; *Phelps et al., 2004*). When extinction is enhanced by replacing aversive with novel, neutral outcomes, the vmPFC was found to be engaged more effectively than during standard extinction (*Dunsmoor et al., 2019*). When extinction is enhanced by replacing aversive outcomes with a reward (counterconditioning), evidence in rodents suggests stronger engagement of the ventral striatum, a region known to be involved in the anticipation and receipt of reward (*Diekhof et al., 2012*). However, human studies only provide indirect evidence for such a mechanism since involvement of the ventral striatum could only be shown during spontaneous recovery (*Keller et al., 2022*) or during reinstatement (*Bulganin et al., 2014*), but not during CC itself. Although it was observed that brain areas of the fear network are reduced during CC versus regular extinction in humans (*Keller et al., 2022*), it is unclear how this difference is achieved. Therefore, although evidence suggests that CC is more effective than regular extinction in preventing the return of threat responses, the neural mechanisms are not well understood yet. It remains unclear whether CC is a form of enhanced extinction that is mediated by enhanced engagement of extinction networks, including the vmPFC, or whether it is driven by engagement of reward networks.

To investigate the qualitative differences between CC versus regular extinction further, category conditioning can be used, a procedure in which conditioned threat responses are learnt by coupling a US to conceptually linked exemplars that together form a category (e.g. pictures of animals)

(*Dunsmoor et al., 2012*). It allows for the typical measures of threat conditioning but also provides the opportunity to probe episodic memory for the CS category exemplars (*Dunsmoor and Kroes, 2019*). Specifically, retrieval of a picture probes the retrieval of specific episodic elements, whereas retrieval of the category-threat association (fear retrieval) takes place at a conceptual level and may be semantic in nature. When episodic memory was probed 24 hr after CC and extinction, it was shown that memory for CS + stimuli that had undergone CC was stronger than memory for CS + stimuli that had undergone regular extinction (*Keller and Dunsmoor, 2020*). This suggests that compared to regular extinction, CC can enhance episodic memory consolidation and potentially provide stronger retrieval competition against a threat memory.

To investigate the neural mechanisms that distinguish CC from regular extinction and to establish whether CC is indeed associated with a memory that is qualitatively different from the safety memory established during regular extinction, we performed a two-day fMRI study comparing CC versus regular extinction in a between-subjects design (*Figure 1A*). Participants underwent category conditioning and subsequently either aversive-to-appetitive CC (CC group) or regular extinction (Ext group; *Figure 1B–C*). During the CC task, participants in the CC group obtained monetary rewards depending on how quickly they responded to a cue superimposed on novel category exemplars from the CS +category, a procedure similar to the monetary incentive delay (MID) task (*Knutson et al., 2000*). To maximize task similarity between tasks and groups, the cued-response element was kept consistent in all tasks (acquisition, CC/extinction, spontaneous recovery, reinstatement), but response-time contingent monetary rewards were only present during the CC task (*Figure 1F*). To assess the potential of CC versus regular extinction in persistently attenuating the expression of threat response, we tested retrieval of the threat memory and reinstatement of threat responses one day later (*Figure 1D*). Episodic memory for exemplars of the CS categories that were presented during threat conditioning and CC/extinction was assessed by means of a surprise memory test. To characterize pupil dilation responses (PDRs) and skin conductance responses (SCRs) during the anticipation of shock- and reward-reinforcement independently from prior conditioning, a separate valence-specific response characterization task was included at the end of the experiment (*Figure 1E*).

In line with previous results (*Kang et al., 2018*, *Keller et al., 2022*), we hypothesized that CC compared to extinction would lead to a more persistent attenuation of threat responses. As indicated above, this could be mediated by two possible neural mechanisms: either through enhanced engagement of extinction networks, reflected by increased engagement of the vmPFC, or through a shift towards reward networks, reflected by activation of the ventral striatum. Based on previous results (*Keller and Dunsmoor, 2020*), we expected stronger episodic memory for CS exemplars presented during CC, whereas regular extinction would not show such a strengthening effect.

## Results

In the valence-specific response characterization task (*Figure 1E*), we observed that both threat and reward-anticipation induced strong arousal-related PDRs and SCRs (see Appendix 1). However, PDRs allowed for a better differentiation of the two compared to the CS- (*Appendix 1—figure 1A*). Therefore, we focused on PDRs in all analyses and refer to Appendix 1 for details on the analysis of SCRs. During the acquisition task, both groups showed comparable and successful acquisition of differential conditioned threat responses (PDR means ± SD: CC CS+=1.085 ± 0.030, CC CS-=1.054 ± 0.033, Ext CS+=1.084 ± 0.050, Ext CS-=1.050 ± 0.035; for PDR, SCR, and fMRI results see Appendix 1).

### Extinction and aversive-to-appetitive counterconditioning

After threat acquisition, participants in the CC group underwent CC, while participants in the Ext group underwent regular extinction (see *Figure 1A* for design overview). Across both groups and phases (early vs. late), we observed retention of conditioned differential PDRs CS-type (CS+, CS-) × Phase (Early, Late) × Group (CC, Ext) rmANOVA, main effect CS-type: $F_{(1,34)}=15.393$, $p<0.001$, $\eta^2=0.312$, *Figure 2A*, as well as a decrease in PDRs over the course of the task (main effect phase: $F_{(1,34)}=10.121$, $p=0.003$, $\eta^2=0.229$). These findings are in contrast to our expectation of a CS-type × Phase × Group interaction. Specifically, we expected differential PDRs to become extinguished in the Ext group, while being sustained in the CC group, potentially due to increased reward anticipation. Extinction in the Ext group, however, already occurred during the early phase (paired t-test, early CS+

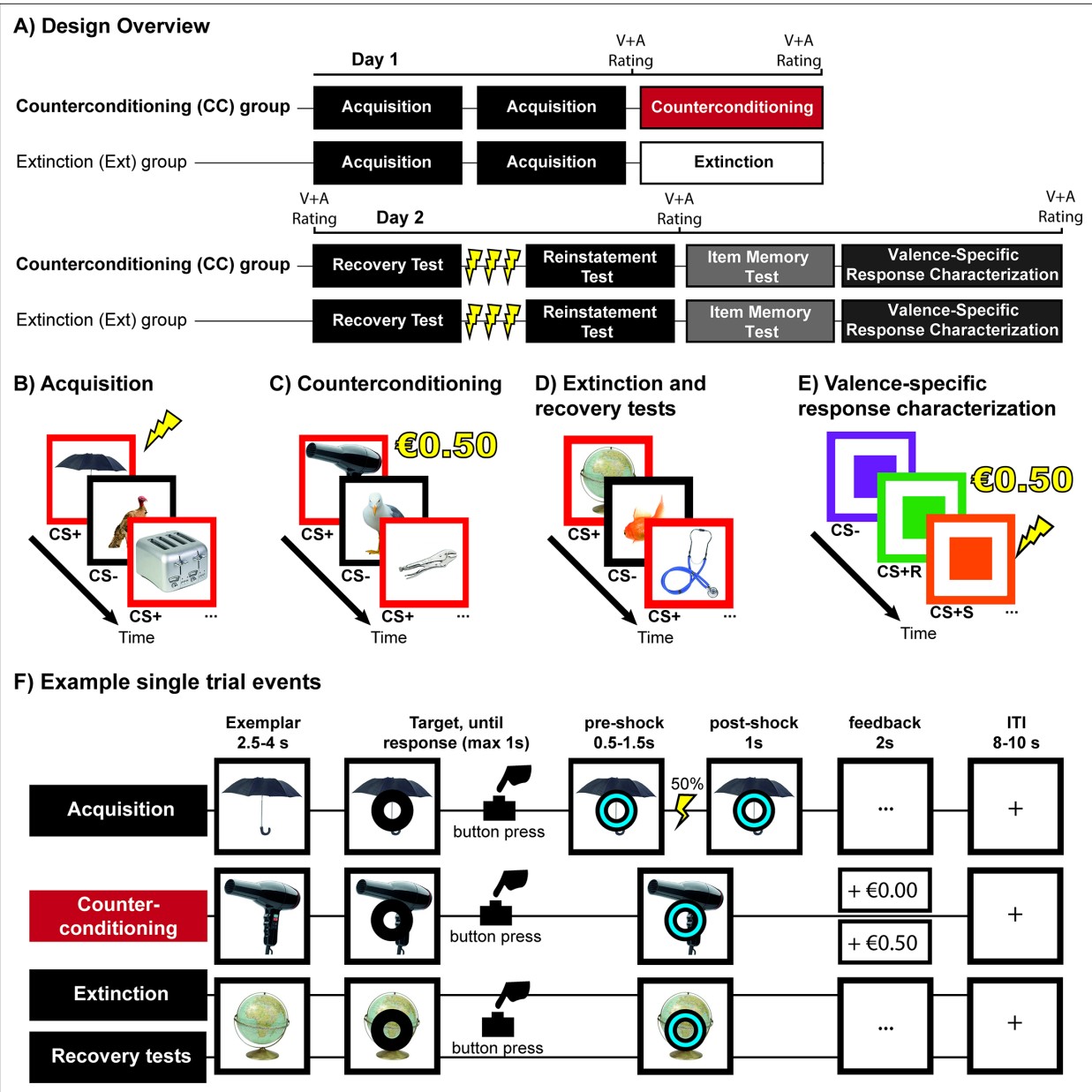

**Figure 1.** Overview of the experimental design. (**A**). Participants were assigned to the counterconditioning (CC) or extinction (Ext) group. On day 1, participants performed two blocks of acquisition of category-conditioned threat responses separated by a 30 s break, followed by CC or extinction. Day 2 consisted of a spontaneous recovery test, a reinstatement procedure and test, an item memory test and a valence-specific response characterization. Valence and arousal ratings for the different categories were taken before or after the tasks as indicated by 'V+A Rating.' All tasks were performed in an MRI scanner. (**B**) During acquisition, participants viewed trial-unique exemplars of objects and animals. Exemplars of one category (CS + animals or objects counterbalanced) were paired with a shock in 50% of trials. CS- trials were not reinforced. (**C**) Participants in the CC group could earn a monetary reward if they responded quickly enough to exemplars in the CS + category. (**D**) Participants in the Ext group underwent extinction. During the extinction task, recovery test,f and reinstatement test, neither CS + nor CS- exemplars were paired with a shock. (**E**) In the valence-specific response characterization task, participants viewed three different colored squares. One color was associated with shock (CS +S), one color with reward (CS +R), and one color served as CS-. The trial structure was otherwise identical to the acquisition and CC tasks. (**F**) In all Pavlovian tasks, trial onset was marked by presentation of a unique category exemplar. After a variable interval, a ring appeared, to which participants were instructed to respond as quickly as possible. Upon response, the ring shifted in color as response confirmation. In the acquisition task, shocks could occur 0.5–1.5 s after the response window had elapsed (indicated as 'pre-shock'). The category exemplar and cue remained visible 1 s after potential shock administration (indicated as 'post-shock'). During CC, participants received visual feedback for 2 s (+€0.50 approximately the fastest 70% of trials, +€0.00 on other trials). During the other tasks, participants viewed neutral feedback (three dots). Trials were separated by an 8–10 s intertrial interval, during which a fixation cross is displayed in the centre of the screen.

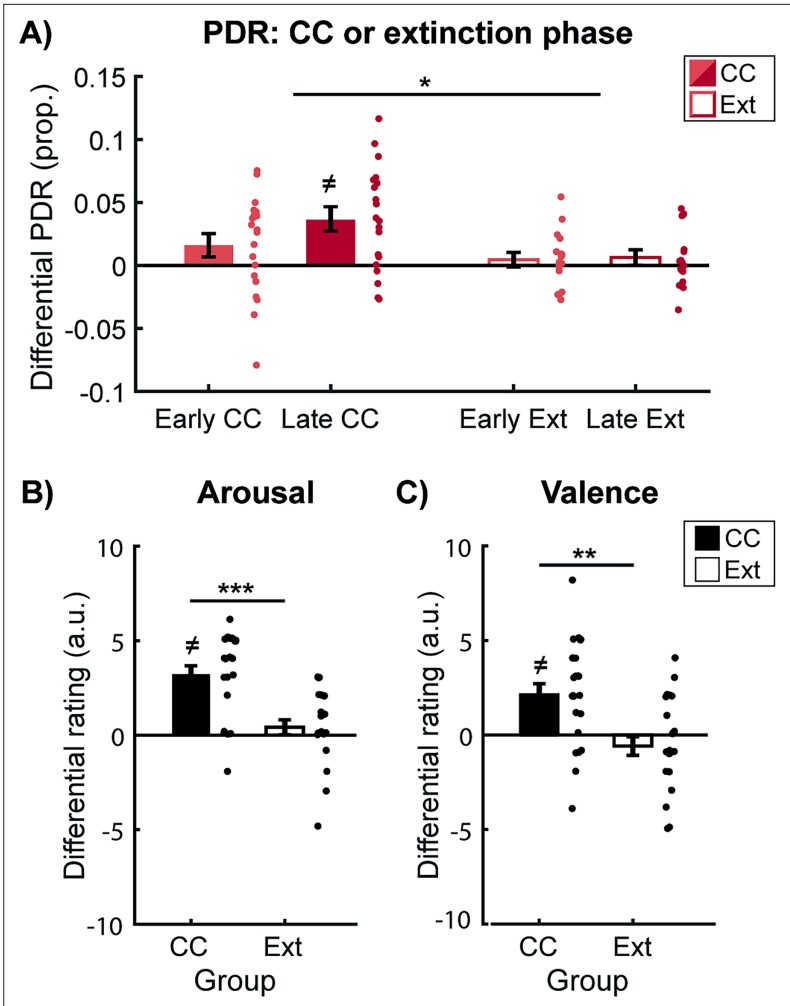

**Figure 2.** Differential PDRs during counterconditioning (CC)/extinction and explicit ratings of arousal and valence provided after the counterconditioning or extinction phase. (**A**) Differential pupil dilation responses (PDRs) for the early (light red) and late (dark red) phase of counterconditioning (CC, solid bars) or extinction (EXT, open bars). Participants undergoing CC showed increased differential PDRs as compared to participants undergoing extinction (CS-type x Group interaction, N=36). (**B**) Arousal and (**C**) valence ratings displayed separately for participants assigned to the counterconditioning (CC, solid bars) and extinction (EXT, open bars) groups. Participants who had undergone CC gave stronger differential arousal scorings than participants who had undergone extinction (CS-type x Group interaction, N=46). In addition, participants who underwent CC showed flipped differential valence ratings: while valence differential valence ratings were negative after extinction, the direction reversed to positive differential ratings after CC (CS-type x Group interaction, N=46). Error bars represent ± standard error of the mean. *=p<0.05, **=p<0.01, ***=p<0.001, ≠ indicates that the bar is significantly different from 0.

vs CS-, *p*=0.233), and differential responses did not change towards the late phase (*p*=0.979). As a result, we found distinct differential conditioned PDRs throughout the CC/extinction task between groups (CS-type × Group interaction: F(1,34)=6.053, *p*=0.019, $\eta^2$=0.151), with participants undergoing CC showing stronger PDRs to CS+ vs CS- category exemplars (paired t-test average CS+ vs CS-, t(20)=3.602, *p*=0.002, CS+: 1.07±0.04, CS-: 1.04±0.04), whereas differential PDRs were extinguished in participants undergoing extinction (paired t-test average CS+ vs CS-, *p*=0.246, CS+: 1.05±0.04, CS-: 1.04±0.04). Results of the valence-specific response characterization task showed that differential PDRs can also be indicative of anticipation of reward (*Appendix 1—figure 1A*). Thus, while PDRs in the Ext group indicated that differential conditioned threat responses were successfully extinguished, differential PDRs persisted in the CC group, likely reflecting reward anticipation. Differential SCRs persisted during the late phase of both CC and extinction but were no longer detectable in the last two trials and were comparable between groups (see Appendix 1).

Valence and arousal ratings provide further support for the extinction of differential responses in the Ext group and positive, reward-induced arousal for CS+ items in the CC group (*Figure 2B–C*). Differential valence ratings for the CS +and CS- differed between groups after the CC/extinction task (CS-type (CS+, CS-) × Group (CC, Ext) rmANOVA, CS-type × Group interaction: F(1,44)=12.054, p=0.001, $\eta$²=0.215). Participants in the CC group rated CS + stimuli more positive than CS- stimuli (t(21)=3.469, p=0.002, CS+: 7.5±0.30, CS-: 5.41±0.38), while participants in the Ext group gave both categories similar valence ratings (p=0.245, CS+: 5.63±0.32, CS-: 6.21±0.28). Since there were group-dependent differences in valence ratings after acquisition (the CS- category was rated more positively by the Ext compared to the CC group, see Appendix 1), we ran an additional analysis, adding differential valence ratings after fear acquisition as a covariate. Results suggested that the group difference in differential valence ratings after CC/extinction remained (main effect Group: F(1,43)=7.364, p=0.010, $\eta$²=0.146). Differential arousal ratings for the CS+ and CS- also differed between groups CS-type (CS+, CS-) × Group (CC, Ext) rmANOVA, CS-type × Group interaction: (F(1,44)=20.862, p<0.001, $\eta$²=0.322). Participants in the CC group reported higher arousal levels for the CS+ category than for the CS- category (t(21)=6.370, p<0.001, CS+: 6.64±0.20, CS-: 3.45±0.38) while participants in the Ext group gave similar arousal ratings for the CS+ and CS- categories (p=0.290, CS+: 4.21±0.43, CS-: 3.80±0.40). Taken together, more positive valence and higher arousal ratings for the CS+ in the CC group as compared to the Ext group further support the interpretation of increased differential PDRs reflecting arousal induced by reward anticipation.

## CC prevents differential spontaneous recovery

To investigate whether CC prevented the spontaneous recovery of differential conditioned threat responses one day later (see *Figure 1A* for design overview), we compared PDRs in the last two trials of the CC/extinction task and the first two trials of the spontaneous recovery test in a CS-type (CS+, CS-) × Group (CC, Ext) × Phase (last two trials of CC/extinction, first two trials of the spontaneous recovery test) rmANOVA [16 participants were excluded due to (partially) missing data, leaving 15 participants in both groups (total N=30)]. We expected the Ext group to show an increase in PDRs

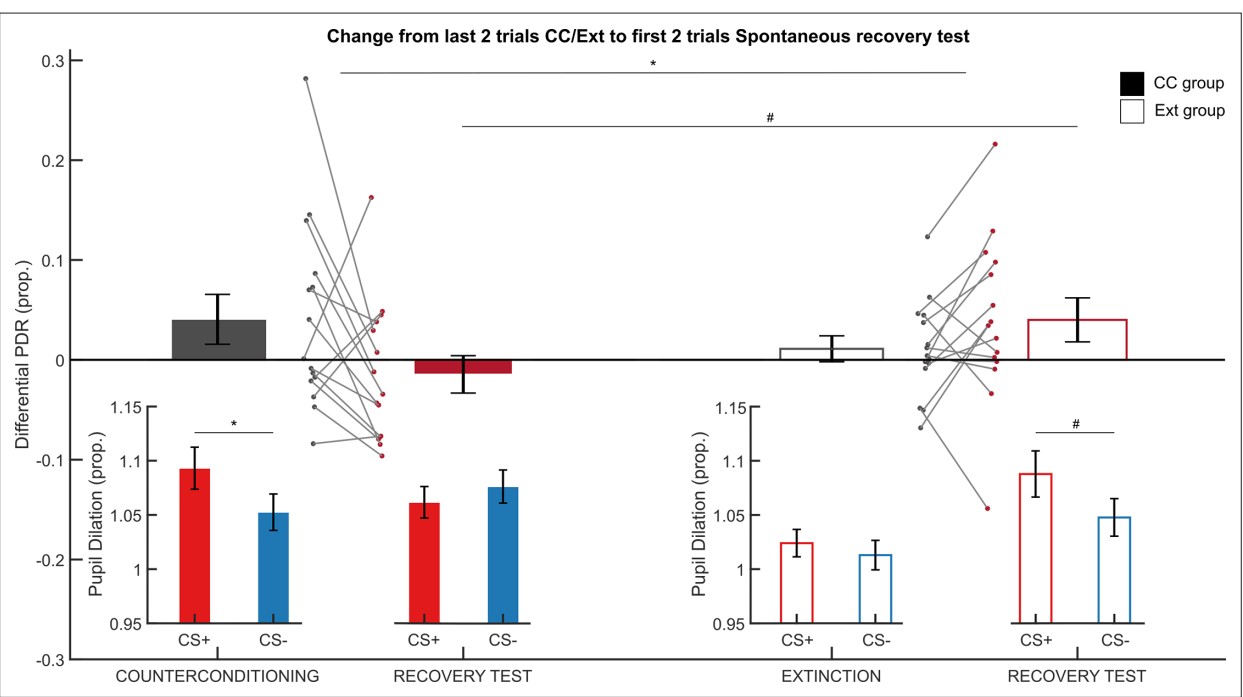

**Figure 3.** Differential pupil dilation responses (PDRs) during the last two trials of counterconditioning or extinction and the first two trials of the spontaneous recovery test. Differential PDRs show selective spontaneous recovery after extinction (Ext, open bars) but not after counterconditioning (CC, solid bars). During the first two trials of the spontaneous recovery test, differential PDRs are increased in the Ext group as compared to the CC group (CS-type x Group interaction, N=31). Insets show PDRs to the CS+ (red) and CS- (blue) during the last two trials of CC/Ext and the first two trials of the spontaneous recovery test. While the Ext group shows differential responding during the spontaneous recovery test (paired t-test, N=15), the CC group does not (paired t-test, N=16). Error bars represent ± standard error of the mean. *=p<0.05, #=p<0.05 one-tailed significance.

from the extinction task to the spontaneous recovery task, while we expected PDRs for the CC group to remain stable or decrease. Critically, differential spontaneous recovery of PDRs differed between groups (Group × CS-type × Phase interaction: F(1,28)=6.329, *p*<0.018, $\eta$²=0.184, *Figure 3*). While the CC group showed a decrease in differential PDRs from CC to spontaneous recovery (t(14)=-1.807, *p*=0.046, one-tailed, CC: 0.34±0.2, spontaneous recovery: –0.01±0.18), the Ext group showed an increase in differential PDRs (t(14)=1.850, *p*=0.043, one-tailed significance, extinction: 0.11±0.01, spontaneous recovery: 0.04±0.02). To conclude, while we observed differential spontaneous recovery in the Ext group, we did not find evidence for differential spontaneous recovery in the CC group, suggesting that CC attenuated the recovery of threat responses compared to regular extinction.

However, since participants undergoing CC showed persistent differential PDRs during the last two trials of the CC phase (likely due to reward anticipation), while participants undergoing extinction did not, we additionally explored whether there was differential responding during the first two trials of the spontaneous recovery test. During the first two trials of the spontaneous recovery test, participants in the CC group showed decreased differential PDRs as compared to the Ext group (CS-type (CS+, CS-) x Group (CC, Ext) rmANOVA, CS-type × Group interaction: F(1,29)=3.901, *p*=0.029, one-tailed, $\eta$²=0.119). Further exploration within the groups confirmed that participants in the CC group did not show retention of differential responses (paired t-test, CS+ and CS- responses during the first two trials of the spontaneous recovery test, *p*=0.219, one-tailed), while the Ext group did show increased responses to the CS+ as compared to the CS- (t(14)=1.958, *p*=0.035, one-tailed). Thus, both the differential spontaneous recovery of PDRs between sessions and differential responding within the first two trials of the spontaneous recovery test suggested that CC prevented spontaneous recovery of differential responses compared to extinction. SCRs did not show differential recovery and were comparable between groups (see Appendix 1).

CC also appeared to have lasting beneficial effects on valence ratings compared to extinction. At the start of the second testing day, differential valence ratings continued to differ between groups (CS-type (CS+, CS-) × Group (CC, Ext) rmANOVA, CS-type × Group interaction: F($_{1,44}$)=5.160, *p*=0.028, $\eta$²=0.105). While participants in the CC group gave similar valence ratings to both categories (*p*=0.179, CS+: 6.3±0.34, CS-: 5.4±0.35), participants in the Ext group gave more negative valence ratings to the CS + category than to the CS- category (t(23)=-1.964, *p*=0.031 one-tailed test, CS+: 5.5±0.30, CS-: 6.3±0.24), also illustrative of relapse of threat associations. Since the CC group received no rewards during the spontaneous recovery task and was aware of this, it was expected that the effect of reward anticipation on PDRs would be weakened in the CC group. So while PDRs did not differ between CS+ and CS- in the CC group because no rewards were given, CS + and CS-items were still rated of similar valence. In comparison, the Ext group rated the CS+ significantly more negative and threat responses to the CS + returned.

While participants in the CC group showed heightened differential arousal ratings immediately after CC as compared to ratings from participants who had undergone extinction (*Figure 2B*), participants in both groups gave comparable differential arousal ratings at the start of the second day immediately before the spontaneous recovery test (CS-type (CS+, CS-) × Group (CC, Ext) rmANOVA, main effect of CS-type: F(1,44)=10.932, *p*=0.002, $\eta$²=0.022, CS+: 4.8±0.28, CS-: 3.9±0.24). Likewise, response times to the CS + and CS- during the first two trials of the spontaneous recovery task were similar across both groups (all p's>0.2). These findings may suggest that differential arousal evoked by the categories was similar in both groups immediately before and during the spontaneous recovery test.

The spontaneous recovery test was followed by a reinstatement procedure, consisting of three unsignaled shocks, and a reinstatement test. However, mean PDRs decreased from spontaneous recovery to reinstatement (t(30)=3.063, *p*=0.005, last two trials of spontaneous recovery: 1.04±0.01, first two trials of reinstatement: 1.01±0.01). Given that we did not observe successful reinstatement in either group, our reinstatement test was not informative on whether CC can lead to a more persistent attenuation of threat responses as compared to regular extinction. A full description of PDR and SCR results of the reinstatement test can be found in Appendix 1.

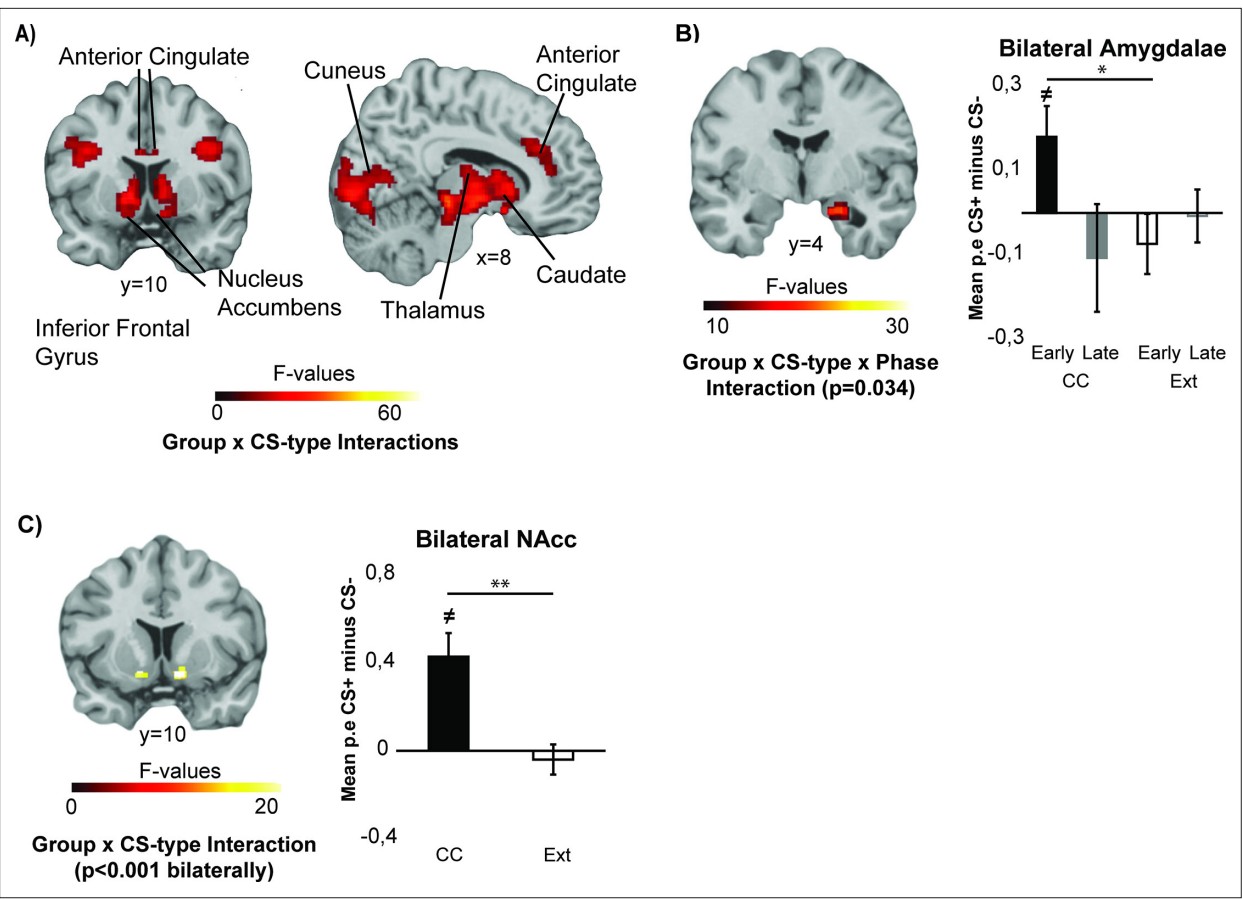

**Figure 4.** Stimulus-type specific activation differs between participants undergoing counterconditioning (CC) versus extinction. (**A**) Whole-brain Group × CS-type interaction effects (N=46) revealed distinct stimulus-specific activation of regions, including the anterior cingulate, cuneus, nucleus accumbens, caudate, thalamus, and inferior frontal gyrus during the counterconditioning vs. extinction phase. Panel A displays group F-images (see **Table 1** for directions) FWE-corrected at *p*<0.05, cluster-forming threshold *p*=0.001. (**B**) The right amygdala showed a Group × CS-type × Phase interaction (N=46) during the CC/extinction task, indicating that CC compared to extinction is associated with decreased activation of the amygdala. (**C**) The bilateral nucleus accumbens (NAcc) showed a Group × CS-type interaction (N=46) during the CC/extinction task, revealing increased NAcc activation in response to the CS + compared to the CS- in the CC but not in the extinction (Ext) group. Panel B and C display group F-images FWE-SVC at *p*<0.05, cluster-forming threshold *p*=0.001, along with post-hoc tests on mean parameter estimates from the complete ROI included in the analyses. \*\**p*<0.01, \**p*<0.05, ≠ indicates that the value is significantly different from 0.

## Distinct CS-type specific activation for extinction and appetitive counterconditioning

During the CC/extinction task, whole-brain analysis revealed that CS-type-specific activation changed differentially between the two groups in a large cluster encompassing multiple regions in the medial temporal lobe (Group × CS-type × Phase interaction, cluster size = 1760 mm$^3$, *p*=0.034, whole-brain FWE-corrected, **Figure 4B** and **Table 1**). We further investigated the anatomical location of the cluster using our ROIs to probe for activity and found that the effect encompassed the amygdala. To further investigate the interaction effect in the amygdala, we extracted parameter estimates from the complete bilateral amygdalae (Automated Anatomic Labeling, AAL, atlas in the WFU PickAtlas toolbox in MN152 space) and performed post-hoc comparisons. In the early phase, CS-type specific responses differed between the groups (t(1,44)=2.173, *p*=0.035, CC: 0.18±0.08, Ext: –0.073±0.08). Specifically, the CC group showed increased amygdala activation to the CS+ as compared to the CS- (t(23)=2.210, *p*=0.037) while that was not the case in the Ext group (*p*=0.390). In the late phase, differential responses were comparable between the groups (*p*=0.503).

Whole-brain analysis further revealed a number of clusters showing distinct CS-specific activations between groups throughout the task, including the anterior cingulate, cuneus, nucleus accumbens, caudate, thalamus, and inferior frontal gyrus (**Figure 4A**, **Table 1**). The group and stimulus-specific

**Table 1.** Whole-brain main effects of group (counterconditioning CC, extinction Ext), CS type (CS+, CS-), and phase (early, late) and interactions, during the counterconditioning/extinction task.

Cluster-forming threshold *p*=0.001, FWE-corrected at *p*<0.05, clusters were labeled using the Talairach Daemon atlas and the automated anatomic labeling (AAL) atlas for ROIs. For each cluster, the peak voxel coordinates (Montreal Neurological Institute, MNI space) and regions are reported, and additional regions contained within the cluster are added in italics. See *Appendix 1—table 1* for main effects of CS-type.

**Peak MNI coordinate**

| Region | Cluster | x | y | z | Size (mm3) | pFWE (cluster) | Peak F-value | Direction |
|---|---|---|---|---|---|---|---|---|
| **Group × CS-type × phase** | | | | | | | | |
| Parahippocampal Gyrus BA34R<br>*Parahippocampal Gyrus Amygdala, Uncus BA34R* | 1 | 18 | -8 | –20 | 1760 | 0.034 | 23.40 | CS +>CS- difference increases from early to late phase for CC, not for Ext |
| **Group × CS-type** | | | | | | | | |
| *Lateral Geniculum Body LR, Caudate Head LR, Thalamus LR, Lentiform Nucleus LR* | 1 | 2 | –26 | –18 | 29920 | <0.001 | 73.15 | (CC CS +>CS-) > (Ext CS +>CS-) |
| Cuneus L<br>*Lingual Gyrus BA17/BA18 LR, Posterior Cingulate LR, Cuneus BA18R, Cuneus BA30L Declive R* | 2 | -6 | –96 | 2 | 23272 | <0.001 | 43.50 | |
| Inferior Frontal Gyrus BA47L<br>*Insula BA13 L* | 3 | –36 | 18 | -6 | 4504 | 0.009 | 30.62 | |
| Extra-Nucleus R | 4 | 30 | 26 | 2 | 3136 | 0.016 | 37.67 | |
| Superior Temporal Gyrus L<br>*Superior Temporal Gyrus BA41 L, Transverse Temporal Gyrus L* | 5 | –60 | –44 | 14 | 9088 | 0.002 | 43.56 | |
| Transverse Temporal Gyrus BA41 R<br>*Superior Temporal Gyrus R, Superior Temporal Gyrus BA42/BA22R* | 6 | 44 | –22 | 12 | 7784 | 0.003 | 42.17 | |
| Anterior Cingulate BA32R<br>*Anterior Cingulate BA32L, Cingulate Gyrus R* | 7 | 6 | 30 | 26 | 8880 | 0.002 | 27.90 | |
| Precentral Gyrus L<br>*Inferior Frontal Gyrus L* | 8 | –36 | 0 | 30 | 3624 | 0.014 | 30.10 | |
| Precentral Gyrus R<br>*Sub-Gyral R* | 9 | 40 | 2 | 32 | 4056 | 0.011 | 40.64 | |
| Precentral Gyrus BA6L<br>*Middle Frontal Gyrus BA6L* | 10 | –44 | -6 | 52 | 2184 | 0.028 | 24.34 | (CC CS +>CS-) > (Ext CS +>CS-) |
| Angular Gyrus R<br>*Supramarginal Gyrus R* | 11 | 54 | –60 | 36 | 1944 | 0.032 | 24.18 | |
| **Group × Phase** | | | | | | | | |
| *No significant clusters* | | | | | | | | |
| **CS-type × Phase** | | | | | | | | |
| *No significant clusters* | | | | | | | | |
| **Group** | | | | | | | | |
| *No significant clusters* | | | | | | | | |
| **Phase** | | | | | | | | |

*Table 1 continued on next page*

*Table 1 continued*

**Peak MNI coordinate**

| | | | | | | | | |
|---|---|---|---|---|---|---|---|---|
| Inferior Frontal Gyrus R<br>*Inferior Frontal Gyrus BA45 R* | 1 | 30 | 26 | 8 | 4848 | 0.006 | 40.27 | Early Phase >Late Phase |
| Insula L<br>*Superior Temporal Gyrus BA22, Precentral Gyrus L* | 2 | −28 | 26 | 0 | 4368 | 0.007 | 38.41 | |
| Postcentral Gyrus L | 3 | −54 | −24 | 22 | 1768 | 0.031 | 23.75 | |

activation of the NAcc was in line with a priori expectations for the CC phase (*Figure 4C*). To further explore this effect, averaged parameter estimates from the bilateral NAcc ROI (mask acquired from the IBASPM 71 atlas in the WFU PickAtlas toolbox in MNI152 space) were extracted. Across the bilateral NAcc, differential activation was increased in the CC as compared to the Ext group (t(44)=2.731, p=0.009, CC: 0.37±0.10, Ext: 0.04±0.06), with the CC showing increased NAcc activation to the CS +compared to the CS- (t(23)=6.194, p<0.001, CS+: 0.59±1.12, CS-: 0.16±0.09) whereas the Ext group did not (p=0.574).

Contrast estimates in further a priori defined ROIs during the CC/extinction task were submitted to a Group (CC, Ext) × CS-type (CS+, CS-) × Phase (early, late) rmANOVA (*Figure 5*). The bilateral hippocampi (right hippocampus cluster size: 664 mm$^3$, p=0.001, FWE-SVC, left hippocampus cluster size: 112 mm$^3$, p=0.024, FWE-SVC) and the left vmPFC (mask defined as bilateral gyrus rectus and medial orbital gyri, cluster size = 160 mm$^3$, p=0.013, FWE-SVC) showed differentially changing CS-type-specific activations between the groups (Group × CS-type × Phase interaction). While CS+-specific suppression of these regions appeared to increase during the CC task, this was not the case during the extinction task. Post-hoc comparisons on averaged parameter estimates in the bilateral hippocampi confirmed that stimulus-specific suppression increased during the course of the task in the CC group (t(23)=3.280, p=0.003, early CS+-CS-: 0.054±0.07, late: −0.150±0.07), but not in the Ext group (p=0.266). Post-hoc comparisons across the vmPFC ROI also revealed increased CS+-specific suppression in the CC group compared to the Ext group (t(44)=2.221, p=0.032, CC: −0.189±0.06, Ext: −0.070±0.10). While the extinction group showed increased CS+-specific activation from the early to the late phase of the extinction task (t(21)=2.235, p=0.036, early CS+: −0.149±0.08, late CS+: 0.040±0.09), the CC group did not (p=0.120). During the late phase, the CC group showed increased vmPFC deactivation to the CS+ compared to the CS- (t(23)=3.174, p=0.004, late CS+: −0.284±0.06, late CS-: −0.095±0.05), while the Ext group did not (p=0.503). Thus, across both the hippocampus and the vmPFC, CC induced increased stimulus-specific suppression.

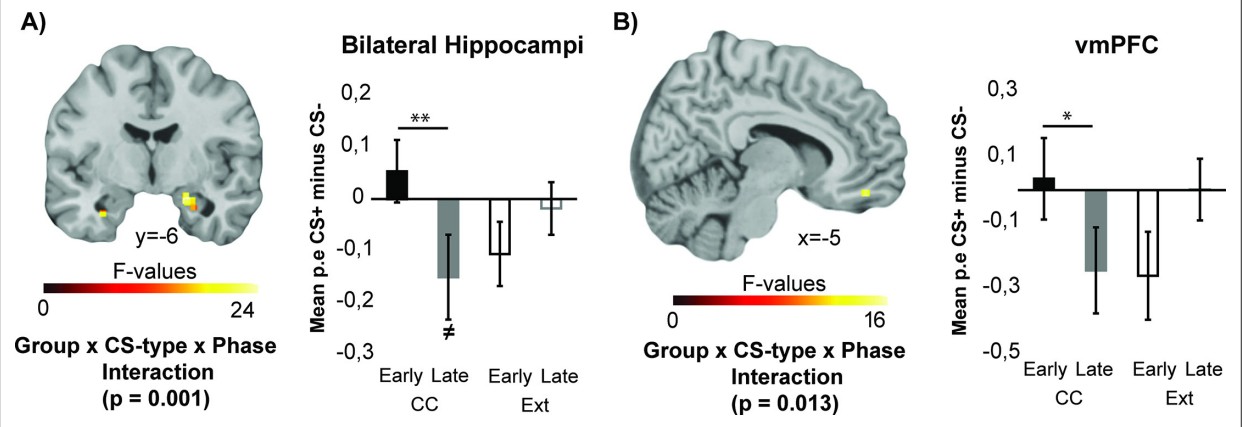

**Figure 5.** ROI analyses during the counterconditioning (CC)/extinction task reveal distinct activity in the hippocampus and left ventromedial prefrontal cortex (vmPFC). During the CC/extinction task, stimulus-specific activation of the hippocampus (**C**) and left vmPFC (**D**) changes differently between groups (N=46). **p<0.01, *p<0.05, ≠ indicates that the bar is significantly different from 0. Group F-images FWE-SVC at p<0.05, cluster-forming threshold p=0.001, along with post-hoc tests on mean parameter estimates from complete ROI included in the analyses.

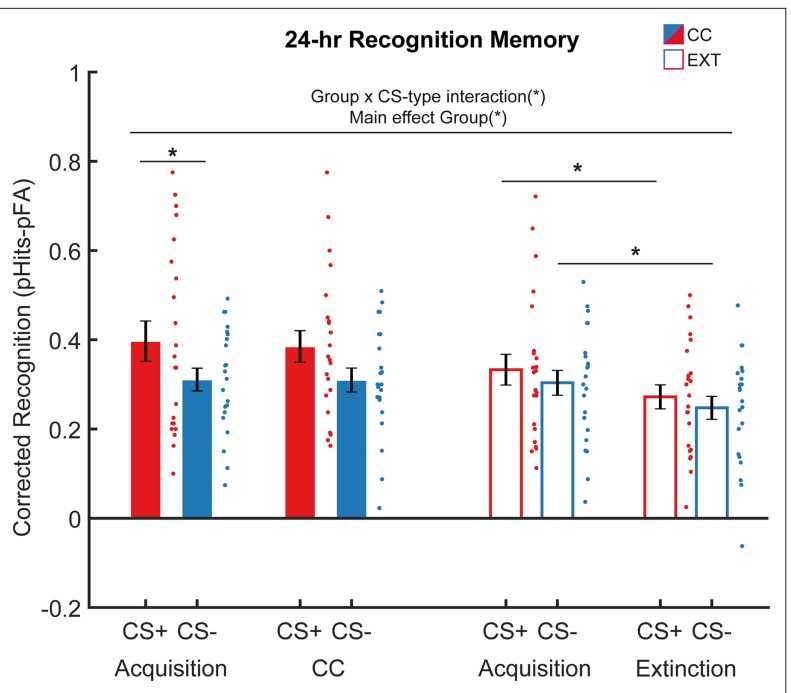

**Figure 6.** Twenty-four-hour recognition memory results. During acquisition and extinction on the first day of the experiment, participants viewed trial-unique exemplars from two semantic categories (objects, animals) that served as CS + and CS-. The next day, participants completed a surprise memory test for these items, mixed with an equal number of novel exemplars. Participants recognized relatively more items from the CS + category (main effect CS-type, N=45), and participants that underwent counterconditioning (CC) showed improved item recognition compared to participants in the extinction (Ext) group (CS-type x Group interaction, N=45). Error bars represent ± standard error of the mean. *=p<0.05.

During the spontaneous recovery task, a priori defined regions of interest did not reveal any effects (see Appendix 1).

## Counterconditioning retroactively enhances item recognition for conditioned exemplars

Following the reinstatement test, participants completed a surprise item recognition test approximately 24 hr after acquisition and the CC/extinction task. One outlier was excluded from this analysis (CS- false alarm rate = 0.91). Threat conditioning has previously been shown to enhance 24 hr item recognition for category exemplars presented during the acquisition phase (**Dunsmoor et al., 2012**). However, this enhancement for CS+ items did not extend to items presented during an extinction session separated from the acquisition phase by a short break (**Dunsmoor et al., 2018**). We, therefore, analyzed item recognition for the CS+ and CS- during acquisition and the CC/extinction phase separately to examine whether the groups differed in recognition memory performance (**Figure 6**).

Corrected recognition scores (hits probability-false alarms probability) were subjected to a task (acquisition, CC/extinction task) × CS-type (CS+, CS-) × Group (CC, Ext) rmANOVA, including CS+-category (animals, tools) as covariate. Overall, participants showed better memory for items from the CS + category (main effect of CS-type: $F(1,42)=10.615$, $p=0.002$, $\eta^2=0.202$) and participants who underwent CC showed better memory as compared to participants who underwent extinction (main effect of Group: $F(1,42)=4.963$, $p=0.031$, $\eta^2=0.106$). Stimulus-type specific item recognition differed between the CC and Ext groups (CS-type × Group interaction: $F(1,42)=4.535$, $p=0.039$, $\eta^2=0.094$). While participants in the CC group showed better recognition memory for the CS+ category compared to the CS- category ($t(22)=2.531$, $p=0.019$, means ± SD: CS+ 0.39 ± 0.17, CS- 0.31±0.10), this was not the case for participants in the Ext group ($t(23)=0.889$, $p=0.384$, means ± SD: CS+ 0.30±0.13, CS- 0.28±0.11). Although the effect of stimulus type was stronger for tools as CS+, this was not different between groups (see Appendix 1). Thus, across the acquisition and CC/extinction phase, participants

who underwent CC showed a stronger enhancement of CS+ memory compared to the participants who underwent extinction.

To further investigate to what extent CC retroactively affected memory for items presented during the acquisition task, we examined item recognition during acquisition and the CC/extinction tasks separately. While threat conditioning increased memory for CS+ items presented during the acquisition task across both groups (main effect CS-type: F(1,42)=18.147, p=<0.001, $\eta$²=0.302), subsequent CC enhanced this effect (Group x CS-type interaction: F(1,42)=5.112, p=0.029, $\eta$²=0.109). Post-hoc tests revealed increased item memory for the CS+ category compared to the CS- category presented during acquisition in the CC group (t(22)=2.341, p=0.029, means ± SD: CS+ 0.40±0.21, CS- 0.31±0.12) but not in the Ext group (t(23)=0.818, p=0.422, means ± SD: CS+ 0.33±0.16, CS-0.30±0.13). Again, although the effect of stimulus type was stronger for tools as CS+, this was not different between groups (see Appendix 1). As the acquisition task was identical between groups, it appears that CC, in comparison to extinction, retroactively enhanced memory for CS + items. For items presented during the CC/extinction task, overall item recognition was better in the CC group compared to the Ext group (main effect group: F(1,42)=8.706, p=0.005, $\eta$²=0.172, means ± SD: CC 0.35±0.12, Ext 0.26±0.09). Thus, compared to regular extinction, CC enhanced recognition of items presented during CC, but interestingly also strengthened the emotional memory enhancement of CS+ exemplars presented during acquisition, suggesting that immediate CC may alter consolidation of a prior threat learning episode.

Following previous work (**Keller and Dunsmoor, 2020**; **Dunsmoor et al., 2018**; **Dunsmoor et al., 2015**), we explored stimulus-type specific decreases in item recognition between tasks, as well as within-phase differences between item recognition for the CS+ and CS- within each group. As expected, a post-hoc paired samples t-test showed that participants in the Ext group remembered significantly more CS+ items from the acquisition phase as compared to the extinction phase (t(23)=2.238, p=0.036, means ± SD: acquisition 0.33±0.16, extinction 0.27±0.13). In contrast, participants who had undergone CC remembered CS+ items presented during acquisition and CC equally well (t(22)=0.390, p=0.701, means ± SD: acquisition 0.40±0.21, CC 0.38±0.16). Thus, while recognition memory for items encoded during the extinction task was substantially weaker than memory for items from the acquisition task, this was not the case for items presented during CC.

## Discussion

This study aimed to test whether CC compared to regular extinction can lead to a more persistent attenuation of threat responses, and to investigate whether this is mediated by neural mechanisms reflecting extinction-related enhanced engagement of the vmPFC or engagement of reward-focused networks. We found that CC prevented differential spontaneous recovery of PDRs compared to regular extinction, suggesting that CC reduces the recovery of threat responses. Notably, there are individual differences (some participants in both groups show the opposite pattern), which should be further investigated in future studies with larger sample sizes, as it is crucial to identify who will respond to treatments based on the principles of standard extinction or counterconditioning. Our fMRI results suggested that CC engages different neural mechanisms compared to extinction. Most notably, while the extinction group showed an increase in CS+-specific vmPFC activation during extinction, the CC group showed CS+-specific deactivation of the vmPFC that persisted throughout the late phase of CC. Furthermore, CC led to increased NAcc activation for the CS + compared to the CS-, whereas this was not the case for extinction. Lastly, phase- and stimulus-specific activation of the hippocampus and the amygdala differed between extinction and CC. Compared to extinction, CC led to increased activation of the amygdala in the early phase and increasing stimulus-specific deactivation of the hippocampus over the course of the early and late phases. In addition, CC retroactively enhanced item recognition for conditioned exemplars presented during acquisition and strengthened memory for conditioned exemplars presented during CC compared to extinction.

The mechanism underlying CC appears to be qualitatively different from the mechanism underlying regular extinction. Regular extinction, which has been interpreted as an implicit form of emotion regulation (**Hartley and Phelps, 2010**), is associated with activation of the vmPFC (**Phelps et al., 2004**, **Milad et al., 2007**), which is thought to inhibit the expression of threat responses by suppressing amygdala activity (**Quirk et al., 2000**; **Morgan et al., 1993**; **Quirk et al., 2003**; **Phelps et al., 2004**). In comparison to regular extinction, novelty-facilitated extinction, a form of enhanced extinction in

which aversive events are replaced with novel neutral outcomes, has shown stronger CS+-specific vmPFC activation (*Dunsmoor et al., 2019*). If CC was similarly mediated by enhanced recruitment of extinction networks, we would have expected increased activation of the vmPFC, yet we observed a CS+-specific deactivation of the vmPFC during CC, disproving this hypothesis. Interestingly, deactivation of the vmPFC during CC was also found in studies investigating a form of counterconditioning induced by means of real-time fMRI decoded neurofeedback (*Koizumi et al., 2017*, *Taschereau-Dumouchel et al., 2018*). During neurofeedback CC, participants implicitly learned to obtain monetary rewards by generating a representation of the target CS + in the visual cortex (*Koizumi et al., 2017*). After neurofeedback CC, reductions in threat responses were stronger in participants showing stronger vmPFC deactivation, suggesting that vmPFC disengagement may be associated with fear reductions (*Koizumi et al., 2017*). Taken together, both our findings and previous neurofeedback studies suggest that, in contrast to enhanced extinction, CC disengages the vmPFC. Given that we replicate this finding using a different approach that includes direct exposure to the CS+, vmPFC disengagement may be a distinguishing characteristic of CC, which could indicate that CC, in contrast to regular extinction, does not involve an implicit regulation strategy. The observed pattern of activity, including vmPFC deactivation, further bears resemblance to activity patterns observed during goal-directed eye movements in an experimental model of eye-movement desensitization and reprocessing (EMDR), which has also been shown to improve extinction learning (*de Voogd et al., 2018*). A similar activity pattern and effect has also been found for working memory-like tasks, such as a game of Tetris (*Holmes et al., 2009*; *James et al., 2015*; *Price et al., 2013*). Given that the above-mentioned tasks associated with vmPFC deactivation share their strong engagement of working memory and/ or endogenous attention mechanisms, thereby engaging the executive control network, deactivation of the vmPFC and hippocampus could be the result of a deactivated default mode network due to competition between activation of large scale brain networks (*Qin et al., 2009*; *Seeley et al., 2007*; *Liang et al., 2016*).

The CC procedure led to clear CS+-specific activation of the NAcc, which is in line with expectations for reward anticipation in tasks with a monetary incentive delay aspect (*Knutson and Cooper, 2005*). Activation of the ventral striatum has also been reported for active avoidance and may be generally associated with instrumental actions as opposed to passive delivery of an outcome (*Boeke et al., 2017*, *Delgado et al., 2009*). In line with studies on active avoidance, delivery of a reward contingent on instrumental actions has been shown to yield CC that is more resistant to renewal (*Thomas et al., 2012*). CS+-specific activation of the NAcc was not seen in participants undergoing extinction, suggesting that this activation is specific to CC. However, previous work in rodents revealed an amygdala-ventral striatum (NAcc) pathway that is activated during extinction training (*Correia et al., 2016*). The recruitment of this pathway was shown to be enhanced during CC and reduce the return of fear (*Correia et al., 2016*), suggesting that CC may, in fact, enhance activation of reward-related networks that are weakly activated by extinction. Indeed, fMRI studies in humans that modeled prediction error for omitted aversive outcomes during extinction training (i.e. outcomes 'better-than-expected') showed involvement of the NAcc (*Raczka et al., 2011*; *Thiele et al., 2021*; *Esser et al., 2021*). Possibly, activation of the NAcc during extinction is limited to early extinction trials generating prediction errors. Nevertheless, based on our findings, it appears that sustained CS+-specific activation of the NAcc is a distinct mechanism underlying CC but not extinction, which is potentially associated with instrumental actions. In comparison to more implicit forms of emotion regulation, as is the case in regular extinction, CC may thus be a more active coping strategy, which is more effective in persistently preventing the return of threat responses (*Boeke et al., 2017*).

A recent neuroimaging study suggests that the neural differences between regular extinction and CC may be maintained over time (*Keller et al., 2022*). In their within-subject study, two CS+ categories (animals, objects) were used during threat conditioning. Subsequently, one of the CS+ categories was used for regular extinction, whereas the other was used for CC. During CC, CS+ exemplars were paired with positively valenced pictures. During a spontaneous recovery task the following day, it was shown that involvement of the vmPFC (amygdala-vmPFC functional connectivity) was stronger for regular extinction compared to CC. In contrast, CS+-specific increases in functional connectivity between the amygdala and the ventral striatum (NAcc) were only observed in the CC condition during a spontaneous recovery task. Both findings are in line with the CC-associated vmPFC deactivation and

NAcc activation that we observed and suggest that differences in the neural mechanisms of regular extinction and CC may be maintained during threat retrieval.

CC compared to regular extinction also strengthened item memory for the conditioned category. While both reward and threat conditioning can enhance item recognition for the CS + category (*Dunsmoor et al., 2012*; *Patil et al., 2017*), recognition of CS + exemplars presented during extinction was shown to drop compared to acquisition (*Dunsmoor et al., 2018*). In contrast to extinction, within-session CC was previously shown to enhance memory, suggesting that CC has a unique, strengthening effect on memory (*Keller and Dunsmoor, 2020*). In the current study, we replicate this finding, showing strengthened memory after CC compared to extinction. While enhanced recognition of items presented during CC could be mediated by attentional prioritization (*Talmi et al., 2008*), CC also retrospectively strengthened memory for items presented during acquisition, suggesting that CC may alter the consolidation of a prior threat conditioning episode. Retroactive enhancement of memory consolidation for related items has previously been shown for conceptually related neutral items presented prior to threat conditioning (*Dunsmoor et al., 2015*) and reward conditioning (*Patil et al., 2017*). At a neurobiological level, these findings have been related to the synaptic tagging-and-capture hypothesis postulating that memories for neutral events can be strengthened if they are followed by salient events, due to an initially short-lived synaptic 'tag' that allows later events to stabilize the memory (*Dunsmoor et al., 2015*, *Ballarini et al., 2009*, *Frey and Morris, 1997*). At a systems level, retroactive memory strengthening has been linked to reverse replay (*Braun et al., 2018*). Specifically, animal research indicates that rewards increase reverse replay (*Ambrose et al., 2016*; *Foster and Wilson, 2006*; *Diba and Buzsáki, 2007*), and reward-induced reverse replay occurs concurrently with firing of midbrain dopamine neurons (*Gomperts et al., 2015*). Interestingly, spontaneous replay is also involved in regular extinction, in which unexpected omission of the US drives spontaneous reactivation of activity patterns in the vmPFC. This spontaneous reactivation was shown to be predictive of extinction recall and could be amplified through pharmacological enhancement of dopaminergic activity (*Gerlicher et al., 2018*). Yet while physiological dopaminergic modulation during extinction may be limited to prediction error signals during the early phase (*Raczka et al., 2011*; *Thiele et al., 2021*; *Esser et al., 2021*), dopaminergic modulation may be sustained throughout the MID-based CC task applied in this study. While we did not measure dopaminergic activity directly, activation of the NAcc during reward anticipation is predictive of dopamine release within the NAcc (*Weiland et al., 2017*; *Weiland et al., 2014*; *Schott et al., 2008*; *Buckholtz et al., 2010*). Given the increased stimulus-specific activation of the NAcc in the CC group, it is likely that dopaminergic activity was enhanced during CC compared to regular extinction. The enhanced dopaminergic modulation could strengthen memories through replay (*Ambrose et al., 2016*, *Singer and Frank, 2009*), or may increase synaptic plasticity directly, potentially explaining enhanced item recognition after CC compared to regular extinction (*Braun et al., 2018*, *Atherton et al., 2015*, *Brzosko et al., 2015*). In line with these findings, research in humans shows that reward systematically modulates memory for neutral objects in a retroactive manner, with objects closest to the reward being prioritized (*Braun et al., 2018*). It could be that reward conditioning during CC similarly drives reward-driven reverse replay, enhancing episodic memory for conceptually related items presented during the preceding acquisition task.

Several limitations of the current study are worth considering. First, while the monetary incentive aspect during CC clearly induced positive valence, it also increased physiological arousal, making it difficult to isolate the individual effects of positive valence and reward-induced arousal. While the current results are in line with previous work in CC using low-arousal, positive-valence pictures (*Keller and Dunsmoor, 2020*), we cannot exclude the possibility that the current findings (in part) reflect differences in task engagement between participants due to active instead of passive reward delivery. However, it is questionable whether it is meaningful to tease individual effects of valence and arousal apart since arousal may facilitate reward processing. Indeed, striatal responses to obtained monetary rewards are dependent on salience and are increased when rewards are dependent on active responses compared to passive delivery (*Zink et al., 2004*). Second, although we included a reinstatement procedure in the experiment, neither the Ext nor the CC group showed differential reinstatement. It is worth noting, however, that reinstatement paradigms in humans may not reliably produce differential reinstatement after extinction (*Haaker et al., 2014*). Third, it is important to note that CC/extinction was carried out within minutes after the acquisition phase, and the effects of CC and

extinction may differ when carried out after the acquisition memory has been consolidated (*Chang and Maren, 2009*; *Devenport, 1998*; *Maren, 2014*; *Myers et al., 2006*). Fourth, whole-brain analysis of the CS-specific activation during the spontaneous recovery test in the Ext group did not yield any clusters above threshold, while physiological results indicated spontaneous recovery of differential threat responses. Given that recovered threat responses are often quick to extinguish and fMRI analyses require averaging across multiple trials to achieve sufficient signal-to-noise ratio, threat-evoked neural activity may have been too brief to be detected. Lastly, there was an unequal sex distribution in our sample, and the sample size did not allow for the investigation of sex-dependent differences, which should be addressed in future studies.

In conclusion, our findings show that appetitive CC improves the retention of safety memory over standard extinction. Strikingly, in contrast to activation of the vmPFC during extinction, CC was associated with stimulus-specific deactivation of the vmPFC. These findings may inform the development of future treatments for fear- and anxiety disorders. While a large body of research focuses on enhancing regular extinction, this study indicates that another promising and potentially longer-lasting approach may be to engage reward circuits. Although further work is needed, a major advantage of CC-based interventions over extinction-based interventions may be that CC could be more tolerable as it may shift attention away from the experience of fear.

## Materials and methods

### Participants

Forty-eight healthy right-handed volunteers (15 males, 33 females; age [22.71±0.44]) with no neurological or psychiatric history, and with normal hearing and normal or corrected-to-normal vision completed the study. Exclusion criteria were pregnancy, disorders of the autonomic system, heart conditions, recreational drug use and any contraindications for MRI. Participants provided written informed consent and were paid 55 euros for their participation. Participants in the CC group were able to earn an additional 14 euros. This study was approved by the local ethical review board (METC Oost-Nederland and CMO Radboudumc). Participants were excluded from the threat acquisition, CC/extinction, spontaneous recovery, and reinstatement analyses if there was no evidence for successful threat acquisition (mean CS->CS + or CS +=CS-). For SCRs, this was the case for three participants; for PDR, this was the case for two participants. Additional participants were excluded in case of (partially) missing data due to technical failure (no data could be recorded or >50% missing data) or in case participants did not return to the second session. These exclusion criteria were preregistered.

### Design and procedure

This study was a two-day between-subjects experiment carried out in the fMRI scanner (see *Figure 1* for an overview of the design). Participants were assigned to either the CC or Ext group according to a predetermined allocation sequence. At the start of each session, two Ag/AgCl electrodes were attached to the medial phalanges of the second and third digits of the left hand, a pulse oximeter was attached to the first digit of the left hand to measure finger pulse and a respiration belt was placed around the abdomen to measure respiration. All measures were taken using a BrainAmp MR system and recorded using the BrainVision Recorder software (Brain Products GmbH, Munich, Germany). The first day consisted of individual adjustment of the electrical shock followed by a single fMRI session that included the following tasks: an object localizer task (17 min), a category threat conditioning task (23 min) and a CC or extinction task (23 min). The second session took place the following day and consisted of three runs: the spontaneous recovery and reinstatement test (12 min), item recognition test (29 min), and the valence-specific response characterization task (17 min).

### Pavlovian conditioning paradigm

Note that CC included an instrumental and not Pavlovian conditioning procedure. This was done because of pragmatic constraints in studies with humans. For example, we cannot food-deprive humans to make an appetitive reward truly reinforcing and make participants anticipate the reward. Previous work by *Patil et al., 2017*, *Zink et al., 2004* and our pilot studies indicated that to maximize reward anticipation and evoke conditioned responses, the reward conditioning needed to be instrumental.

The acquisition, counterconditioning, extinction, spontaneous recovery, and reinstatement tasks consisted of a categorical differential delay threat conditioning paradigm (*Dunsmoor et al., 2012* with elements of the monetary incentive delay task *Knutson et al., 2000*). Participants viewed trial-unique exemplars of pictures from two categories (animals or objects, see *Figure 1*). In a counter-balanced manner, exemplars from one category served as CS+ (reinforced) stimuli, while exemplars from the other category served as CS- (unreinforced stimuli). Each trial started with the presentation of a stimulus. After a variable delay of 2.5–4 s, a cue appeared to which participants were instructed to respond as quickly as possible with a button press. After the button press, or when a 1 s response window had elapsed, the color of the cue shifted from black to blue. 0.5–1.5 s after the response window elapsed, CS+ items presented during the acquisition phase could be reinforced with a shock. During the acquisition phase, 50% of the CS+ pictures were followed by a shock. After 1 s, the stimulus was replaced by neutral feedback during the acquisition, extinction, and recovery tasks. During the CC phase, neutral feedback was replaced by monetary feedback. During the CC phase, participants could obtain a €0.50 reward for their quickest responses to the cues presented on top of CS+ stimuli. The response time target was dynamically adjusted to achieve a reward reinforcement rate of approximately 70%. Reward was withheld during the first three CS+ trials during the CC phase to make the transition from the acquisition to the CC phase more gradual. The inter-trial interval (ITI) varied randomly between 8 and 10 s. Pictures were presented in a pseudorandom order with no more than three consecutive presentations of items from the same category, and CC blocks consisted of 40 CS+ and 40 CS- presentations each. The spontaneous recovery block consisted of 15 CS+ and 15 CS + presentations, and the reinstatement test consisted of 5 CS+ and 5 CS- presentations.

## Item recognition memory test

Participants carried out a surprise recognition memory test comprised of 160 pictures (80 CS+, 80 CS-) shown during the acquisition and CC/extinction phases, as well as 160 category-matched new items (80 CS+, 80 CS-). Participants rated on a 6-point scale whether the picture was 'definitely old,' 'probably old,' 'maybe old,' 'maybe new,' 'probably new,' and 'definitely new'.

## Valence-specific response characterization

The valence-specific response characterization task consisted of an adapted version of the conditioning paradigm used during the acquisition phase. Instead of category items, participants were presented with squares in three different colors. One of the stimuli was reinforced with shocks (CS+-shock, 50% reinforcement rate), one stimulus was reinforced with monetary rewards (CS+-reward, approximately 70% reinforcement rate, response time target adjusted dynamically) and the last stimulus was not reinforced (CS-). Each stimulus was presented 40 times in a pseudorandom order with no more than three repetitions of each stimulus. Colors and reinforcement (shocks vs. rewards) were counterbalanced across participants.

## Peripheral stimulation

Electrical shocks were delivered using two Ag/AgCl electrodes attached to the medial phalanges of the second and third digit of the right hand using a MAXTENS 2000 (Bio-Protech) device. Shock intensity varied in 10 intensity steps between 0–40 V and 0–80 mA. Shock duration was 200ms. In line with prior threat conditioning protocols, shock intensity was calibrated using an ascending staircase procedure starting with a low voltage setting near a perceptible threshold and increasing to a level deemed 'maximally uncomfortable but not painful' by the participant (*Dunsmoor et al., 2015*, *Kroes et al., 2017*, *LaBar et al., 1998*).

## Arousal and valence ratings

Arousal and valence ratings were acquired using self-assessment manikin scales. The arousal scale ranged from 1 (=extremely calm) to 10 (=extremely excited). The valence scale ranged from 1 (=extremely negative) to 10 (=extremely positive). The valence and arousal ratings were collected for the two categories (animals and tools) after the acquisition phase, after the CC/extinction phase, at the start of day 2, immediately before the spontaneous recovery test, and after the reinstatement test. For the stimuli used in the valence-specific response characterization task, valence, and arousal ratings were collected immediately after the task.

## SCR pre-processing and analysis

Electrodermal activity data were pre-processed using in-house software; radio frequency (RF) artefacts were removed and a low-pass filter was applied (*de Voogd et al., 2016b*; *de Voogd et al., 2016a*). Skin conductance responses (SCR) were automatically scored with additional, blinded, manual super-vision using Autonomate (*Green et al., 2014*). SCR amplitudes (measured in µSiem) were determined for each trial as the maximum response with an onset between 0.5 and 7.5 s after stimulus onset and maximum rise time of 14.5 s. Shock- and reward- reinforced trials were excluded from analysis. All response amplitudes were square-root transformed and normalized according to each participant's mean UCS response prior to statistical analysis. The average SCRs were computed per CS-type, task, phase (early, late), and participant.

## PDR pre-processing and analysis

Pupil dilation was measured with a MR-compatible eye-tracker from SensoMotoric Instrument (MEye Track-LR camera unit, SMI, SensoMotoric Instruments) and sampled at a rate of 50 Hz. Data were analyzed using in-house software (*Hermans et al., 2013*) implemented in Matlab R2018b (Math-Works, RRID:SCR_001622), based on previously described methods (*Siegle et al., 2003*). Eyeblink artifacts were identified and linearly interpolated 100ms before and 100ms after each identified blink. Data from scan runs missing 50% time points or more were excluded. After interpolating missing values, time series were band-pass filtered at 0.05–5 Hz (by subtracting the mean and dividing by the standard deviation) within each participant and run to account for between-subjects variance in overall pupil size. Event-related pupil diameter responses were calculated by averaging pupil diameter during the 3.5–7 s period after stimulus onset, divided by the 1 s pre-stimulus pupil diameter (−1–0 s). The average PDRs were computed per CS-type, task, phase (early, late), and participant.

## MRI data acquisition

MRI scans were acquired using a Siemens (Erlangen, Germany) 3T MAGNETOM PrismaFit MR scanner equipped with 32-channel transmit-receiver head coil. The manufacturer's automatic 3D-shimming procedure was performed at the beginning of each experiment. Participants were placed in a light head restraint within the scanner to limit head movements during acquisition. Functional images were acquired with multi-band multi-echo gradient echo-planar (EPI) sequence [51 oblique transverse slices; slice thickness, 2.5 mm; TR, 1.5 s; flip angle, 75°; echo times, 13.4, 34.8, and 56.2 ms; FOV, 210×210 mm$^2$; matrix size 84×84×64, fat suppression]. To account for regional variation in susceptibility-induced signal drop-out, voxel-wise weighted sums of all echoes were calculated based on local contrast-to-noise ratio, after which echo series are integrated using PAID weighting (*Poser et al., 2006*). Field maps were acquired (51 oblique transverse slices; slice thickness, 2.5 mm; TR, 0.49 s; TE, 4.92 ms and 7.48 ms; flip angle, 60°; FOV, 210×210 mm$^2$; matrix size 84×84×64) at the start of each session to allow for correction of distortions due to magnetic field inhomogeneity. A high-resolution structural image (1 mm isotropic) was acquired using a T1-weighted 3D magnetization-prepared rapid gradient echo sequence [MP-RAGE; TR, 2300 ms; TE, 3.03 ms; flip angle, 8°; 192 contiguous 1 mm slices; FOV = 256×256 mm$^2$].

## fMRI analysis

Anatomical and functional data were pre-processed using fMRIPrep 20.0.6 (*Esteban et al., 2019*) (RRID:SCR_016216). Functional MRI data were pre-processed in standard stereotactic (MNI152) space. Pulse and respiration data were processed offline using in-house software and visually inspected to remove artefacts and correct peak detection, and corrected pulse and respiration data were used for retrospective image-based correction (RETROICORplus) of physiological noise artefacts in BOLD-fMRI data (*Glover et al., 2000*). Identical transformations were applied to all functional images, which were resliced into 2 mm isotropic voxels. After pre-processing in fMRIPrep, functional images were smoothed with a 6 mm FWHM Gaussian kernel (using SPM12; http://www.fil.ion.ucl.ac.uk/spm; Wellcome Department of Imaging Neuroscience, London, UK).

For the acquisition, extinction/cc and spontaneous recovery phases, BOLD responses to CS+, and CS- during the early phase (first half of the trials) and late phase (second half of the trials) were modeled in four separate regressors using box-car functions. Additionally, during all these phases, target presentation, button press, and shocks were modeled using stick functions, and feedback

presentation and breaks were modeled using box-car functions and included as nuisance regressors. For the category localizer, BOLD responses to animals, objects, and phase-scrambled blocks were modeled in 3 separate regressors using box functions. All first-level models also included six movement parameter regressors (3 translations, 3 rotations) derived from rigid-body motion correction, 25 RETROICOR physiological noise regressors, high-pass filtering (1/128 Hz cut-off), and AR(1) serial correlations correction. First-level contrasts were calculated for early and late CS+ and CS- separately for the acquisition, CC/extinction, and spontaneous recovery phases.

For the acquisition and CC/extinction, first-level contrasts were entered into a second-level Group (extinction, cc) × CS-type (CS+, CS-) × Phase (early, late) mixed factorial model using the Multilevel and Repeated Measures (MRM) toolbox (*McFarquhar et al., 2016*). For the spontaneous recovery test, BOLD responses from the early phase were entered into a second-level Group (extinction, cc) × CS-type (CS+, CS-) mixed factorial model. Thresholding was achieved using nonparametric permutation testing (5000 iterations), with a cluster-setting threshold of $p<0.001$ for whole-brain analysis and familywise error (FWE) correction at $p<0.05$ at cluster-level for whole-brain analysis and voxel-level for ROI analysis (Amygdala, Hippocampus, vmPFC, NAcc). Activations are displayed on the single-subject high-resolution T1 volume provided by the Montreal Neurological Institute (MNI).

## Region of interest definition

Based on a priori hypotheses, results for the amygdala, NAcc, hippocampus, and the ventromedial prefrontal cortex are corrected for reduced search volumes using small volume. Masks were created using the WFU PickAtlas toolbox (*Maldjian et al., 2003*) (RRID:SCR_007378) in combination with the Automated Anatomical Labeling atlas *Tzourio-Mazoyer et al., 2002* for the bilateral amygdala, bilateral hippocampus, and vmPFC (Frontal_Med_orb_L&R and Rectus L&R). The *IBASPM 71* anatomical atlas toolbox (*Alemán-Gómez et al., 2006*) was used to create a mask for the bilateral NAcc.

## Statistical testing

Statistical analyses of behavioral and physiological variables were performed in SPSS (IBM SPSS Statistics Inc, RRID:SCR_002865). Dependent measures were submitted to repeated measures ANOVAs and statistics were Greenhouse-Geisser or Huynh-Feldt corrected for non-sphericity when appropriate. Repeated measures ANOVAs were applied to capture early versus late phases of acquisition and CC/extinction, as well as to compare late CC/extinction (last two trials) compared to early spontaneous recovery (first two trials). We refrained from trial-by-trial analyses in a relatively small sample, as this would have cost too many degrees of freedom and is not expected to provide more information. Significant findings from ANOVAs were followed up by paired and independent samples t-tests and Bonferroni-adjusted where applicable. We report partial eta-squared as a measure of effect size. Means ± s.e.m are provided where relevant unless otherwise indicated.

## Deviations from the pre-registration

The preregistration for this project can be found on OSF (https://osf.io/fbz6n). We pre-registered to sample SCRS in a 0.75 and 3.15 s window after stimulus onset. However, visual inspection of SCR responses during the acquisition phase indicated that response latencies shifted towards the late phase of the trial. We, therefore, opted to use a longer window (0.5 to 7.5 s for stimulus onset) and exclude reinforced trials. The pre-registration erroneously stated that pupil-dilation data would be z-scored and later divided by the pre-stimulus average. PDR data were not z-scored but were only normalized to a 1 sec pre-stimulus baseline. In line with the SCR data, response onset latencies were later than expected. Based on visual inspection of the data from the acquisition phase, we decided to use a window around the expected shock onset: 3.5–7 s after stimulus onset. Reinforced trials were excluded. Results for SCR, retrospective reinforcement estimations and the reinstatement test can be found in Appendix 1. Due to an error in the scripts for the item recognition test, trial-by-trial data were not recorded for the first 12 participants. Therefore, analysis of the memory data focused on averaged data for the early and late phase of acquisition and CC/extinction, leaving out planned change point analyses on bins of four trials.

While we planned to extract a vmPFC mask for ROI analysis based on a [CS->CS+ shock] contrast of BOLD responses during the valence-specific response characterization task to identify 'extinction regions,' this did not yield ventromedial prefrontal clusters that survived correction. Instead, in line

with our other ROIs, we opted to create a mask based on the AAL atlas. Due to time constraints, native-space and functional connectivity analyses were not carried out for this manuscript.

## Acknowledgements

This work was supported by grants awarded to Marijn Kroes (PhD): H2020 Marie-Sklodowska-Curie fellowship, Society in Science - Branco Weiss fellowship.

## Additional information

### Funding

| Funder | Grant reference number | Author |
| --- | --- | --- |
| H2020 Marie Skłodowska-Curie Actions | Grant awarded to Marijn Kroes (not author) | Lisa Wirz |
| Branco Weiss Fellowship – Society in Science | Grant awarded to Marijn Kroes (not author) | Lisa Wirz |

The funders had no role in study design, data collection and interpretation, or the decision to submit the work for publication.

### Author contributions

Lisa Wirz, Formal analysis, Supervision, Validation, Investigation, Visualization, Writing – original draft, Writing – review and editing; Maxime C Houtekamer, Conceptualization, Data curation, Formal analysis, Investigation, Visualization, Methodology, Writing – original draft; Jette de Vos, Data curation, Formal analysis; Joseph E Dunsmoor, Judith Homberg, Marloes JAG Henckens, Conceptualization, Writing – review and editing; Erno Hermans, Conceptualization, Supervision, Writing – review and editing

### Author ORCIDs

Lisa Wirz ⑧ https://orcid.org/0000-0002-2811-4080
Jette de Vos ⑧ https://orcid.org/0000-0001-5966-8257
Joseph E Dunsmoor ⑧ https://orcid.org/0000-0002-5448-6873
Judith Homberg ⑧ https://orcid.org/0000-0002-7621-1010
Marloes JAG Henckens ⑧ https://orcid.org/0000-0002-2375-1611

### Ethics

Human subjects: This study was approved by the local ethical review board (METC Oost-Nederland and CMO Radboudumc). Participants provided written informed consent and consent to publish.

Reviewer #1 (Public review): https://doi.org/10.7554/eLife.101518.3.sa1
Reviewer #2 (Public review): https://doi.org/10.7554/eLife.101518.3.sa2
Reviewer #3 (Public review): https://doi.org/10.7554/eLife.101518.3.sa3
Author response https://doi.org/10.7554/eLife.101518.3.sa4

## Additional files

### Supplementary files

MDAR checklist

### Data availability

The raw, pseudonomized data, as well as the fMRI subject-level contrast files used for group-level analyses are available from the Radboud Data Repository at https://doi.org/10.34973/kves-ee90. The processed data, fMRI group-level output, SPSS mask to replicate the behavioral and physiological data analyses, and scripts to replicate the publication figures are openly available at https://osf.io/mvsq6.

The following datasets were generated:

| Author(s) | Year | Dataset title | Dataset URL | Database and Identifier |
|---|---|---|---|---|
| Wirz LM, Houtekamer MC, de Vos J, Dunsmoor JE, Homberg J, Henckens MJAG, Hermans EJ | 2026 | Unraveling the mechanisms underlying counter conditioning | https://doi.org/10.34973/kves-ee90 | Data Sharing Collection, 10.34973/kves-ee90 |
| Houtekamer M, de Vos J, Henckens M, Homberg J, Dunsmoor JE, Fernandez G, Hermans E, Wirz L | 2026 | Unraveling the neurocognitive mechanisms underlying counter-conditioning in humans | https://osf.io/mvsq6/ | Open Science Framework, mvsq6 |

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

# Appendix 1

## Valence-specific response characterization

At the end of the experiment, participants underwent a simplified version of the main experimental task, in which category exemplars were replaced by colored squares. This task was used to investigate to what extent skin conductance responses (SCRs) and pupil dilation responses (PDRs) can be used to disentangle anticipation of shock and reward. Participants viewed three different coloured squares and learned that one colour was associated with shocks (CS+S), one colour with rewards (CS+R) and one colour served as CS-. The trial structure was otherwise identical to comparable trials from the acquisition and CC phases. At the end of the task, participants were asked to rate the three stimuli on valence and arousal self-assessment manikin scales (Bradley & Lang, 1994).

During this valence-specific response characterization task, we observed habituation in SCRs over the course of the task (CS-type (CS+S, CS+R, CS-) × Phase (early, late) × Group (CC, Ext) rmANOVA, main effect phase: F(1,38)=13.921, $p$=0.001, $\eta^2$=0.268) and different SCR magnitudes for the three different CS-types (main effect CS-type (CS+S, CS+R, CS-): F(2,76)=78.460, $p$<0.001, $\eta^2$=0.674). In addition, habituation depended on CS-type (CS-type × Phase interaction: F$_{(2,76)}$=6.825, $p$=0.002, $\eta^2$=0.152). During the early phase, SCRs in response to the CS+R and the CS- were not distinguishable (t(40)=0.115, $p$=0.909, CS+R: 0.32±0.03, CS-:0.32±0.03), while during the late phase, SCRs to the CS+R were larger than the CS- (t(40)=4.993, $p$<0.001, CS+R: 0.29±0.03, CS-:0.19±0.02). SCRs to the CS+S were consistently larger than SCRs to the CS+R (early: t(41)=9.345, CS+S: 0.62±0.04, $p$<0.001, late: t(40)=5.952, $p$<0.001, CS+S: 0.56±0.04) and the CS- (early: t(40)=10.020, $p$<0.001, late: t(4)=10.122, $p$<0.001). Thus, anticipation of aversive reinforcement (CS+S) led to increased SCRs compared to anticipation of reward (CS+R) and CS- presentation throughout the task. Due to the fact that SCRs performed less well in differentiation between CS+R and CS-, we focused our analyses on PDRs but report SCR results here as well.

We also observed CS-type dependent differences in PDRs CS-type (CS+S, CS+R, CS-) × Phase (early, late) × Group (CC, Ext) rmANOVA, main effect CS-type (CS+S, CS+R, CS-): F$_{(2,68)}$=19.783, $P$<0.001, $\eta^2$=0.368). In comparison to the neutral CS-, both the shock-reinforced CS+ (CS+S) and reward-reinforced CS+ (CS+R) evoked larger PDRs (*Appendix 1—figure 1A* and t(36)=7.071, $p$<0.001 and t(26)=4.900, $P$<0.001, respectively, CS+S: 1.05±0.03, CS+R: 1.04±0.04, CS-: 1.01±0.02). However, reward- and shock anticipation-induced PDRs did not differ statistically (t(36)=1.146, $p$=0.259). While both shock anticipation and reward anticipation led to similar increases in PDRs as compared to the neutral condition, valence and arousal ratings indicated that participants experienced shock and reward trials differently. Specifically, the CS+R was rated more positive than the CS- (t(47)=9.046, $p$<0.001, CS+R: 7.79±0.14, CS-: 5.96±0.16, *Appendix 1—figure 1C*), while the CS+S was rated less positive than the CS- (t(47)=-10.337, $p$<0.001, CS+S: 2.96±0.25). Participants reported increased arousal to both the CS+S and CS+R as compared to the CS- (t(47)=4.666, $p$<0.001 and t(47)=8.897, $p$<0.001, respectively, CS +S: 5.42±0.35, CS+R: 6.31±0.21, CS-: 3.33±0.30, *Appendix 1—figure 1B*). While it was not possible to distinguish PDRs to the CS+S and CS+R, explicit ratings of arousal were marginally increased for the CS+R as compared to the CS+S (t(47)=-2.100, $p$=0.041). In conclusion, the response characterization task showed that while anticipation of reward and shock both generate increased PDRs as compared to the CS-, distinct retrospective valence ratings show the expected directions.

### Threat acquisition

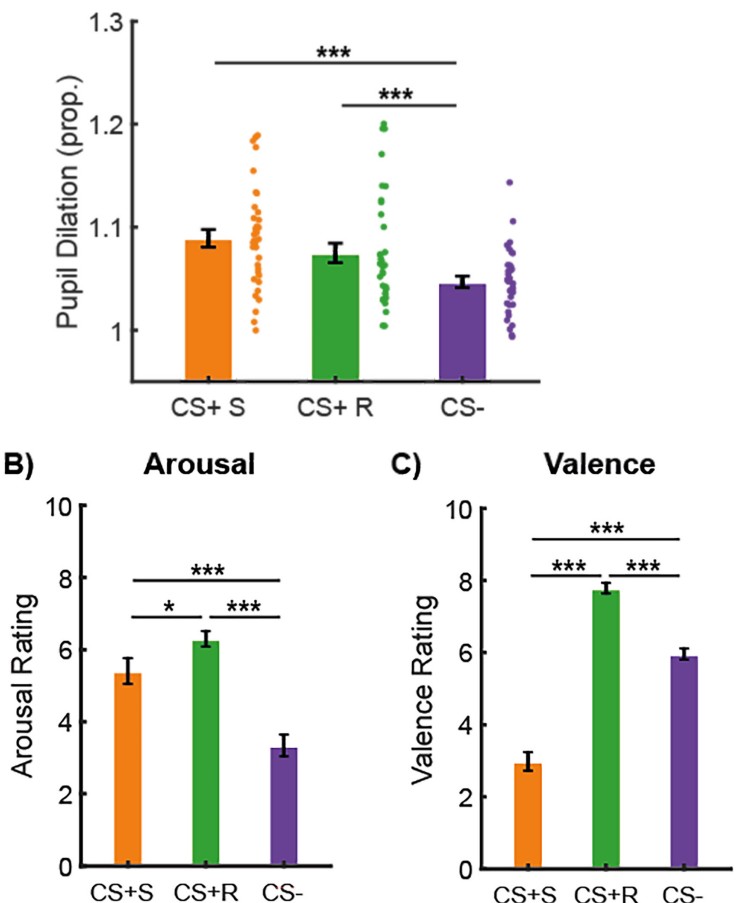

**Appendix 1—figure 1.** Pupil dilation responses (PDRs), explicit arousal, and valence rating for the different CSs presented during the valence-specific response characterization task. (**A**) PDRs to the shock reinforced (CS+S), reward reinforced (CS+R) and CS- stimuli, averaged across the task and all participants. PDRs were increased for the CS+S and CS+R as compared to the CS- (main effect CS-type, N=36) (**B**) Explicit ratings of arousal and (**C**) valence provided immediately after the task. Explicit ratings of arousal for the CS+S exceeded ratings for the CS-, and the CS+R was rated higher in arousal than the CS+S (main effect CS-type, N=47). Valence ratings (1=extremely negative, 10=extremely positive) for the CS+R were more positive than ratings for the CS-, while ratings for the CS+S were more negative than for the CS- and CS+R (main effect CS-type, N=47). Error bars represent ± standard error of the mean *=p<0.05, ***=p<0.001.

### Physiological and behavioral evidence for acquisition of conditioned threat responses

Participants pre-assigned to the CC and Ext groups underwent an identical threat acquisition procedure. To verify that participants pre-assigned to both groups acquired conditioned threat responses of comparable strength, we compared PDRs, explicit valence and arousal ratings, and response times between groups. During the acquisition task, participants pre-assigned to both groups showed stable and comparable differential conditioned threat responses as measured by PDRs (*Appendix 1—figure 2A*, CS-type (CS+, CS-) × Phase (Early, Late) × Group (CC, Ext) rmANOVA, main effect CS-type: $F(1,37)=41.172$, $p<0.001$, $\eta^2=0.533$, other main effects and interactions: all p's>0.2). Both groups also acquired comparable differential SCRs (main effect CS-type: $F_{(1,42)}=58.633$, $p<0.001$, $\eta^2=0.583$), although SCRs showed habituation over the course of the task (main effect phase: $F_{(1,42)}=66.907$, $p<0.001$, $\eta^2=0.614$, all other p's>0.3). Thus, both SCRs and PDRs demonstrated comparable acquisition of conditioned threat responses between groups.

Successful threat acquisition was further confirmed by valence and arousal ratings for the CS+ and CS- categories at the end of the acquisition task. Arousal ratings for the CS+ category exceeded arousal ratings for the CS- category (*Appendix 1—figure 2B*, CS-type (CS+, CS-) × Group (CC, Ext) rmANOVA, main effect CS-type: $F(1,44)=27.573$, $p<0.001$, $\eta^2=0.385$), and did not differ between groups (all p's>0.2). Similarly, the CS+ category was given lower valence (less positive) ratings than the CS- category (*Appendix 1—figure 2C*, CS-type (CS+, CS-) × Group (CC, Ext) rmANOVA, main effect CS-type: $F(1,44)=12.626$, $p<0.001$, $\eta^2=0.223$). Although there was no main effect of group on valence ratings ($p>0.7$), the effect of CS-category unexpectedly differed between the CC and Ext group (CS-type × Group interaction: $F(1,44)=4.512$, $p=0.039$, $\eta^2=0.093$), due to more positive ratings to the CS-category in the Ext group (CC: 5.8±0.4, Ext: 6.9±0.3, $t(44)=2.156$, $p=0.037$). Nevertheless, valence ratings for the CS+ category were comparable between groups (CC: 5.1±0.5, Ext: 4.2±0.4, $p>0.1$), suggesting that the strength of the acquired threat responses is likely similar between groups.

To keep all experimental tasks similar between groups, participants in both groups were asked to respond to targets that were superimposed on the stimuli as quickly as possible. To verify that both groups performed similarly on this task, we compared response times for the different stimuli between the groups. During the acquisition task, participants responded faster to targets in CS+ trials compared to CS- trials (CS-type (CS+, CS-) × Group (CC, Ext) rmANOVA, main effect CS-type: $F(1,45)=10.839$, $p=0.002$, $\eta^2=0.194$), with no differences between groups (all p's>0.058).

After the acquisition task, participants in both groups reported higher estimated reinforcement rates for the CS+ category as compared to the CS- category (CS-type (CS+, CS-) × Group rmANOVA, main effect CS-type: $F(1,45)=82.176$, $p<0.001$, $\eta^2=0.646$). The reported reinforcement rates did not differ between groups (all p's>0.3).

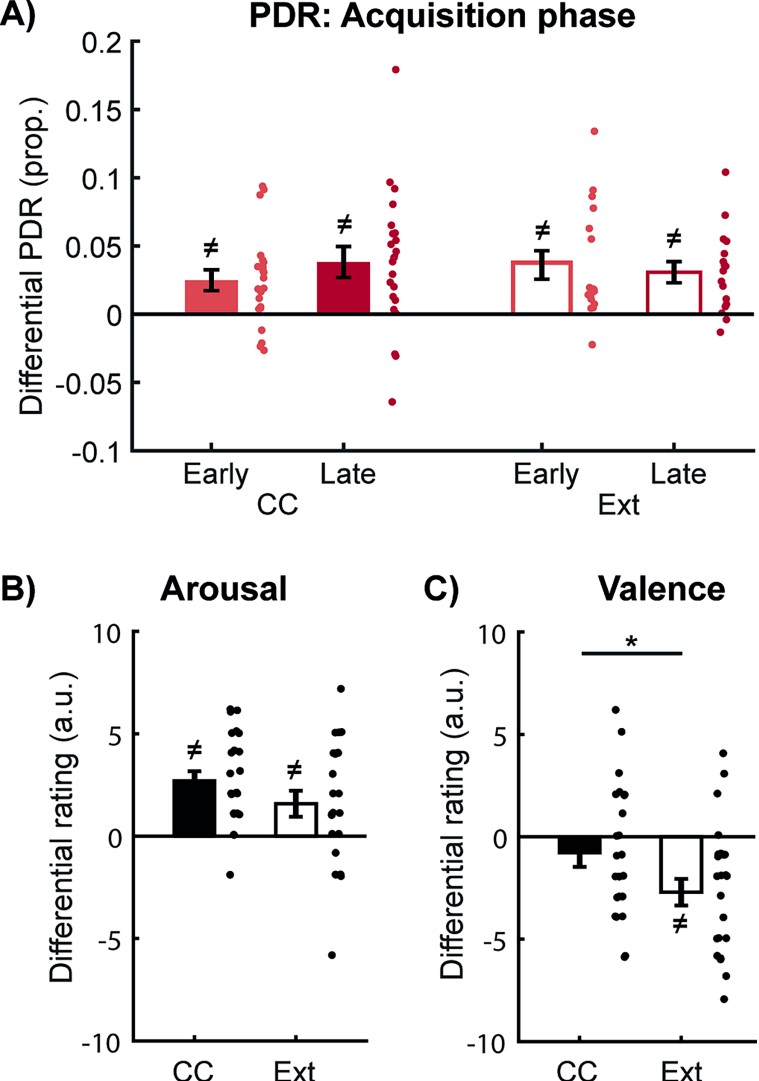

**Appendix 1—figure 2.** Differential pupil dilation responses (PDRs) during acquisition and explicit ratings of arousal and valence provided after acquisition. (**A**) Differential PDRs for the early (light red) and late (dark red) phase of the acquisition task, (**B**) arousal and (**C**) valence ratings, displayed separately for participants assigned to the counterconditioning (CC, solid bars) and extinction (EXT, open bars) groups. Both groups showed comparable differential PDRs and arousal ratings during the acquisition task. Participants in both groups showed negative differential valence ratings (stronger negative valence for CS+ vs CS-), although this effect was stronger in the Ext group (CS-type x Group interaction, N=46). Error bars represent ± standard error of the mean. *$p<0.05$. ≠. Significantly different from 0.

## Brain activation supports successful acquisition of conditioned threat responses

The acquisition of conditioned fear on the first day reliably activated networks associated with fear conditioning. Whole-brain analysis identified regions that were more responsive to the CS+ versus, the CS- category (***Appendix 1—figure 3***, see ***Appendix 1—table 1*** for a complete overview of findings). We observed differential BOLD responses in a large number of brain areas, including the bilateral insula, posterior and anterior cingulate, thalamus, precuneus (undirected test, cluster size = 425400 mm³, $p<0.001$, whole-brain FWE-corrected) and the bilateral amygdala (right cluster size = 1088 mm³, $p<0.001$, FWE-SVC, left cluster size = 736 mm³, $p<0.001$, FWE-SVC).

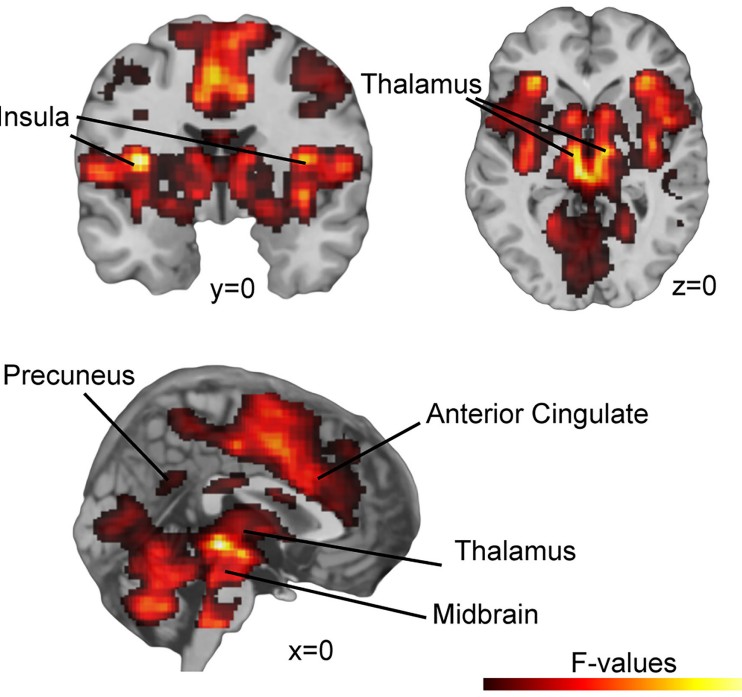

**Appendix 1—figure 3.** Differential threat responses during acquisition revealed CS-specific activation of clusters encompassing a range of regions including the bilateral insula, thalamus, precuneus, anterior cingulate and midbrain. Group F-image of the effect of CS type, thresholded at cluster-level FWE-corrected *p*<0.05, cluster-forming threshold *p*=0.001, displayed on the *single-subject* high-resolution *T1* volume provided by the Montreal Neurological Institute (MNI).

**Appendix 1—table 1.** Whole-brain main effects of group (counterconditioning CC, extinction Ext), CS type (CS+, CS-), and phase (early, late) and interactions, during the acquisition task. Cluster-forming threshold *p*=0.001, FWE-corrected at *p*<0.05, clusters were labeled using the Talairach Daemon atlas and the automated anatomic labeling (AAL) atlas for ROIs. For each cluster, the peak voxel coordinates (Montreal Neurological Institute, MNI space) and regions are reported, and additional regions contained within the cluster are added in italics.

| Region | Cluster | Peak MNI coordinates x | y | z | Size (mm³) | pFWE (cluster) | Peak F-value | Direction |
|---|---|---|---|---|---|---|---|---|
| **CS-type × phase** | | | | | | | | |
| Parahippocampal Gyrus L *Insula L, Parahippocampal Gyrus Hippocampus L, Claustrum L, Lentiform Nucleus Putamen L, Uncus L, Postcentral Gyrus BA43 L* | 1 | −18 | −10 | −16 | 6656 | 0.005 | 25.86 | Early CS+>Late CS+ |
| Parahippocampal Gyrus Amygdala R *Inferior Frontal Gyrus R, Subcallosal Gyrus BA34R* | 2 | 20 | -4 | −22 | 2116 | 0.033 | 25.34 | Early CS+>Late CS+ |
| Culmen L *Declive L, Lingual Gyrus L* | 3 | -8 | −54 | −16 | 1720 | 0.027 | 20.57 | Early CS+>Late CS+ |

*Appendix 1—table 1 Continued on next page*

*Appendix 1—table 1 Continued*

| | | Peak MNI coordinates | | | | | | |
|---|---|---|---|---|---|---|---|---|
| Parahippocampal Gyrus L<br>*Parahippocampal gyrus BA36L/BA30L, Culmen L* | 4 | −20 | −42 | -2 | 5236 | 0.006 | 31.65 | |
| Medial Frontal Gyrus BA11 L<br>*Anterior Cingulate BA32L, Medial Frontal Gyrus BA10 R, BA11 R* | 5 | -4 | 38 | −14 | 1104 | 0.046 | 17.23 | |
| Superior Temporal Gyrus L<br>*Middle Temporal Gyrus BA21/BA22 L* | 6 | −52 | 10 | −14 | 3056 | 0.015 | 28.06 | |
| Lingual Gyrus BA18/BA19 R | 7 | -6 | −68 | -2 | 2584 | 0.018 | 35.21 | |
| Insula R<br>*Inferior Parietal Lobule R, Superior Temporal Gyrus BA22 R, Postcentral Gyrus BA3 R, Superior Temporal Gyrus BA22 R, Precentral Gyrus BA4/BA6, Inferior Parietal Lobule BA40, Middle Temporal Gyrus, Superior Temporal Gyrus BA42* | 8 | 38 | -6 | 18 | 24488 | 0.001 | 27.62 | |
| Parahippocampal Gyrus R | 9 | 24 | −36 | -4 | 1432 | 0.035 | 22.26 | Early CS+>Late CS+ |
| Inferior Frontal Gyrus BA45 R | 10 | 52 | 14 | 14 | 1392 | 0.036 | 21.37 | |
| Precentral Gyrus L | 11 | −60 | -8 | 32 | 5640 | 0.006 | 22.09 | |
| Inferior Parietal Lobule BA40L Postcentral *gyrus BA2L* | 12 | −56 | −36 | 42 | 1104 | 0.046 | 20.42 | |
| Precuneus L<br>*Postcentral gyrus L, Cingulate gyrus L* | 13 | −14 | −42 | 54 | 1496 | 0.034 | 19.85 | |
| Precuneus R<br>*Paracentral Lobule Ba7 R, Precuneus R, Cingulate gyrus R, Superior Parietal Lobule BA7 R* | 14 | 20 | −52 | 54 | 5528 | 0.006 | 24.86 | |
| Medial Frontal gyrus L (23)<br>*Medial frontal gyrus BA6LR, Paracentral Lobule L* | 15 | -6 | −20 | 64 | 1840 | 0.025 | 21.47 | |
| **CS-type** | | | | | | | | |
| Postcentral Gyrus L<br>*Inferior Parietal Lobule LR, Insula LR, Postcentral Gyrus R, Cingulate Gyrus LR, Thalamus LR, Caudate LR, Inferior- Middle- and Superior Frontal Gyrus LR, Posterior Cingulate R, Precentral Gyrus LR, Precuneus L, Delice R, Culmen R, Cuneus L, Superior Temporal Gyrus LR, Anterior Cingulate LR, Parahippocampal Gyrus BA27 R, Lentiform Nucleus LR* | 1 | −50 | −20 | 16 | 425400 | <0.001 | 195.37 | CS +>CS- |
| Posterior Cingulate BA31 L<br>*Precuneus M* | 2 | -4 | −56 | 24 | 2816 | 0.021 | 26.17 | |
| Corpus Callosum M<br>*Corpus Callosum R* | 3 | 0 | 0 | 22 | 1296 | 0.049 | 35.45 | |
| Angular Gyrus R<br>*Angular Gyrus BA39 R, Precuneus R* | 4 | 56 | −66 | 30 | 2432 | 0.024 | 31.42 | CS +<CS- |
| Angular Gyrus BA39L | 5 | −54 | −68 | 30 | 5584 | 0.010 | 36.02 | |
| Superior Frontal Gyrus BA9L<br>*Superior Frontal Gyrus BA8L, Middle Frontal Gyrus BA6L* | 6 | −18 | 40 | 42 | 7200 | 0.007 | 33.18 | |
| **Phase** | | | | | | | | |
| Superior Temporal Gyrus LR,<br>*Inferior Parietal Lobule R, Middle Temporal Gyrus LR, Inferior- Middle- and Superior Frontal Gyrus LR, Caudate LR, Middle Occipital Gyrus LR, Cingulate Gyrus LR, Anterior Cingulate LR, Declive LR, Precuneus LR, Insula LR, Culmen LR, Superior Temporal Gyrus LR, Lingual Gyrus LR, Fusiform Gyrus LR, Angular Gyrus R, Claustrum LR, Thalamus LR, Parahippocampal Gyrus LR, Cuneus LR* | 1 | −64 | −38 | 12 | 784632 | <0.001 | 77.44 | Early >Late |

## Counterconditioning and extinction

### Skin conductance responses

Differential SCRs were still apparent during the CC/extinction task (CS-type (CS+, CS-) × Phase (Early, Late) × Group (CC, Ext) rmANOVA, main effect CS-type: $F(1,40)=17.609$, $p<0.001$, $\eta^2=0.306$). To verify that successful extinction was reached by the end of the task, we explored SCRs in the late phase separately but found that differential SCRs were still apparent in that phase ($F(1,41)=12.166$, $p=0.001$, $\eta^2=0.229$). Finally, we explored whether the last two trials of the extinction task showed evidence of residual differential SCRs. In the last two trials of extinction, across both groups, there was no evidence for differential SCRs (all p's>0.2). Thus, while differential SCRs persisted during the late phase of the extinction task, differential responses were no longer apparent in the last two trials.

Throughout the CC/extinction, there was no evidence for different SCRs between groups, suggesting that participants in both groups underwent comparable but slow extinction of differential SCRs.

## Overlapping stimulus-specific activation during counterconditioning and extinction

A number of clusters showed comparable stimulus-specific activations during CC and extinction (*Appendix 1—table 2*).

**Appendix 1—table 2.** Whole-brain main effect of CS-type during the counterconditioning (CC)/ extinction task.

Cluster-forming threshold *p*=0.001, FWE-corrected at *p*<0.05, clusters were labeled using the Talairach Daemon atlas and the automated anatomic labeling (AAL) atlas for ROIs. For each cluster, the peak voxel coordinates (Montreal Neurological Institute, MNI space) and regions are reported, and additional regions contained within the cluster are added in italics.

**Peak MNI coordinate**

| Region | Cluster | x | y | z | Size (mm³) | pFWE (cluster) | Peak F-value | Direction |
|---|---|---|---|---|---|---|---|---|
| **CS-type** | | | | | | | | |
| Caudate Head L<br>*Thalamus LR, Caudate Head R, Substantia Nigra LR* | 1 | –10 | 10 | -2 | 25136 | 0.001 | 60.98 | CS +>CS- |
| Insula R<br>*Inferior Frontal Gyrus R, Precentral Gyrus BA44 R, Inferior Frontal Gyrus BA45 R* | 3 | 28 | 26 | 0 | 22800 | 0.001 | 89.75 | |
| Inferior Frontal Gyrus L<br>*Insula BA13 L* | 4 | –32 | 28 | 0 | 7808 | 0.004 | 52.01 | |
| Lingual Gyrus L<br>*Inferior Occipital Gyrus L, Cuneus L, Middle Occipital Gyrus L* | 5 | –24 | –80 | –12 | 5696 | 0.006 | 32.81 | |
| Superior Temporal Gyrus R<br>*Transverse Temporal Gyrus R* | 6 | 50 | –18 | 6 | 3744 | 0.012 | 30.65 | |
| Lingual Gyrus R<br>*Cuneus R* | 7 | 8 | –94 | 6 | 3688 | 0.012 | 27.81 | |
| Superior Temporal Gyrus L<br>*Transverse Temporal Gyrus L* | 8 | –44 | –24 | 8 | 3864 | 0.011 | 44.50 | |
| Anterior Cingulate BA32 R<br>*Medial Frontal Gyrus BA8 R, Anterior Cingulate LR, Cingulate Gyrus BA32 R* | 9 | 4 | 38 | 20 | 7064 | 0.004 | 26.61 | |
| Superior Temporal Gyrus R<br>*Supramarginal Gyrus R, Inferior Parietal Lobule BA40R* | 10 | 64 | –34 | 14 | 3720 | 0.012 | 25.88 | |
| Superior Temporal Gyrus L | 11 | –60 | –46 | 16 | 2504 | 0.023 | 36.62 | |
| Cingulate Gyrus L<br>*Posterior Cingulate BA23R, Posterior Cingulate L* | 13 | -6 | –20 | 30 | 3928 | 0.011 | 38.33 | |
| Angular Gyrus L<br>*Middle Temporal Gyrus L, Angular Gyrus BA39 L* | 12 | –44 | –64 | 32 | 4104 | 0.010 | 23.58 | CS->CS + |
| Inferior Temporal Gyrus BA21 L,<br>*Middle Temporal Gyrus BA21 L* | 2 | –64 | –10 | –22 | 1696 | 0.039 | 27.05 | |
| Angular Gyrus R<br>*Supramarginal Gyrus R* | 14 | 44 | –66 | 34 | 1392 | 0.050 | 18.35 | |
| Postcentral Gyrus BA40R<br>*Precentral Gyrus Ba4/BA3 R* | 15 | 34 | –40 | 58 | 1704 | 0.038 | 19.84 | |
| Middle Frontal Gyrus BA8/BA6 L | 16 | –24 | 16 | 48 | 4576 | 0.008 | 34.03 | |

## Spontaneous recovery test

### Skin conductance responses

To investigate whether CC can prevent spontaneous recovery of differential SCRs, SCRs during the last two trials of extinction and the first two trials of the spontaneous recovery test were submitted to a CS-type (CS+, CS-) × Phase (last two trials of the CC/extinction phase, first two trials of the

spontaneous recovery test) × Group (CC, Ext) rmANOVA. SCRs showed a generalized increase from the last two trials of extinction to the first two trials of the spontaneous recovery test (main effect phase: F(1,38)=32.392, p<0.001, $\eta^2$=0.460)). There was evidence for differential SCRs across both phases (main effect CS-type: (F(1,38)=9.560, p=0.004, $\eta^2$=0.201), as CS+ stimuli evoked higher SCRs than CS- stimuli (t(43)=2.518, p=0.016, CS+:0.41±0.03, CS-:0.35±0.03), yet we did not find evidence for CS+-specific spontaneous recovery or effect of group (all p's>0.4). Thus, although there was a generalized increase in responding from the end of extinction to the start of the spontaneous recovery test, SCRs did not show differential recovery and were comparable between groups.

## Brain activation

During the spontaneous recovery test, CS-specific activation differed between groups in the inferior temporal gyrus (cluster size = 2008 mm³, p=0.020, FWE-corrected, *Appendix 1—figure 4A*) and the inferior frontal gyrus (cluster size = 1920 mm³, p=0.022, FWE-corrected, *Appendix 1—figure 4B*). Separate analysis of the spontaneous recovery phase within each group did not reveal any suprathreshold clusters in the Ext group, while a number of clusters showed stimulus-specific activation in the CC group. Specifically, the CC group showed stimulus-specific activation in the bilateral fusiform gyri, superior parietal lobes, and inferior frontal gyri, and in the right thalamus, caudate, middle frontal gyrus and angular gyrus (see *Appendix 1—table 3*). A priori defined regions of interest (ROIs) during the spontaneous recovery task were submitted to a Group (CC, Ext) × CS-type (CS+, CS-) × Phase (early, late) rmANOVA but did not reveal any effects.

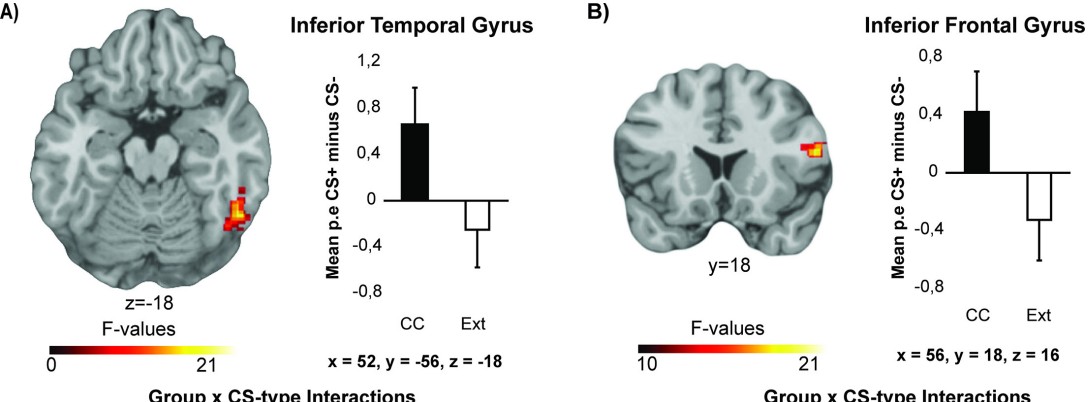

**Appendix 1—figure 4.** During the spontaneous recovery test, stimulus type-specific activation of the inferior temporal and frontal gyri differed between groups. The inferior temporal gyrus (**A**) and inferior frontal gyrus (**B**) show increased CS+-specific activation in the counterconditioning (CC) group as compared to the extinction (Ext) group (main effect group, N=46). Group F images thresholded at FWE-corrected *p*<0.05, cluster-forming threshold *p*=0.001, displayed on the *single-subject* high-resolution *T1* volume provided by the Montreal Neurological Institute (MNI) and parameters estimates from peak voxels.

**Appendix 1—table 3.** Peak voxel coordinates and statistics of activations during the spontaneous recovery phase in the counterconditioning (CC) group.
Clusters were labeled using the automated anatomic labeling (AAL) atlas. For each cluster, the peak voxel coordinates and regions are reported, and additional regions contained within the cluster are added in italics. Clusters are whole-brain FWE-corrected at *p*<0.05.

**Peak MNI coordinate**

| Region | Cluster | x | y | z | Size (mm3) | pFWE (cluster) | Peak T-value | Direction |
|---|---|---|---|---|---|---|---|---|
| CS-type | | | | | | | | |

*Appendix 1—table 3 Continued on next page*

*Appendix 1—table 3 Continued*

**Peak MNI coordinate**

| | | | | | | | | |
|---|---|---|---|---|---|---|---|---|
| Thalamus R<br>*Parahippocampal Gyrus R* | 1 | 10 | –22 | -4 | 2160 | <0.001 | 6.70 | CS +>CS- |
| Inferior temporal Gyrus R<br>*Fusiform gyrus R* | 2 | 46 | –52 | -2 | 4856 | <0.001 | 6.56 | |
| Inferior frontal gyrus, triangular R | 3 | 58 | 26 | 26 | 4992 | <0.001 | 5.93 | |
| Superior parietal lobe R<br>*Angular Gyrus R* | 4 | 26 | –60 | 50 | 2048 | <0.001 | 5.36 | |
| Inferior frontal gyrus, orbital part L | 5 | 50 | 26 | -6 | 1120 | 0.019 | 5.31 | |
| Fusiform gyrus L<br>*Lingual gyrus* | 6 | –38 | –80 | –18 | 1176 | 0.015 | 5.18 | |
| Caudate R | 7 | 6 | 0 | 10 | 1952 | <0.001 | 4.87 | |
| Middle Frontal gyrus R | 8 | 37 | 0 | 44 | 992 | 0.03 | 4.86 | |
| Superior parietal lobule L<br>*Angular gyrus L* | 9 | –32 | –58 | 58 | 1480 | 0.004 | 4.53 | |

## Reinstatement test

SCRs showed a generalized increase from the spontaneous recovery phase to the reinstatement test (CS-type (CS+, CS-) × Group (CC, Ext) × Phase (spontaneous recovery test, reinstatement test) rmANOVA, main effect Phase: $F(1,39)=25.758$, $p<0.001$, $\eta^2=0.398$, last two trials of spontaneous recovery: $0.22\pm0.04$, first two trials of reinstatement: $0.38\pm0.03$). Across the last two trials of the spontaneous recovery test and the first two trials of the reinstatement test, differential SCRs differ between the counterconditioning and extinction group (interaction effect of stimulus type and group: $F_{(1,39)}=4.967$, $p=0.032$, $\eta^2=0.113$). Yet, there is no evidence for differential reinstatement between groups (no CS-type × Phase × Group interaction, $p=0.218$). Moreover, mean SCRs to CS+ and CS- stimuli do not differ within either group (all p's>0.12).

PDRs showed a generalized decrease from the spontaneous recovery phase to the reinstatement test (CS-type (CS+, CS-) × Group (CC, Ext) × Phase (spontaneous recovery test, reinstatement test) rmANOVA, main effect of Phase ($F(1,29)=9.104$, $p=0.005$, $\eta^2=239$)). Mean PDRs decreased from spontaneous recovery to reinstatement ($t(30)=3.063$, $p=0.005$, last two trials of spontaneous recovery: $1.04\pm0.01$, first two trials of reinstatement: $1.01\pm0.01$). Given that we did not observe successful reinstatement in either group, our reinstatement test does not inform us about whether CC can lead to a more persistent attenuation of fear as compared to classic extinction.

## CS+-specific enhancement of recognition memory depends on CS+ category

Corrected recognition scores (pHits – pFA) were subjected to a task (acquisition, CC/extinction task) x CS-type (CS+, CS-) × Group (CC, Ext) rmANOVA, including CS+-category (animals, tools) as covariate. Although the effect of CS-type differed depending on the category used as CS+ (CS-type × CS+-category interaction: $F(1,42)=19.400$, $p<0.001$, $\eta^2=0.316$) and task (CS-type × CS+-category × task interaction: $F(1,43)=5.375$, $p=0.025$, $\eta^2=0.113$), showing a stronger effect for tools as CS+, this did not differ between our experimental groups (CS-type × CS+-category × Group interaction: $F(1,41)=0.050$, $p=0.824$; CS-type × task × CS+-category × Group interaction: $F_{(1,41)}=0.005$, $p=0.946$).

To further investigate to what extent CC retroactively affected memory for items presented during the acquisition task, we examined recognition of items presented during acquisition and the CC/extinction tasks separately. Retrospective memory enhancement for the CS+ items compared to CS- items differed depending on the CS+ category during both the acquisition task (CS-type × CS+ category interaction: $F(1,42)=29.730$, $p<0.001$, $\eta^2=0.414$, CS+ category main effect: $F(1,42)=5.346$, $p=0.026$, $\eta^2=0.113$) and the CC/extinction task (CS-type × CS+ category interaction: $F(1,42)=5.121$, $p=0.029$, $\eta^2=0.109$), showing a stronger effect for tools as CS+. Importantly, this effect was comparable between groups (CS-type × CS+ category × Group interaction: acquisition task $F(1,41)=0.016$, $p=0.806$; CC/extinction task $F(1,41)=0.019$, $p=0.890$).

