## [Editor Report · eLife Assessment]

This **important** work combines self-report, neural and physiology data to examine the efficacy and mechanisms of counter conditioning versus extinction in reducing re-emergence of conditioned threat responses and show that this appears to rely on the nucleus accumbens rather than the ventromedial prefrontal cortex. These findings are supported by **convincing** evidence, though some areas could benefit from a few targeted refinements. The findings will be of interest to researchers across multiple subfields, including neuroscientists, cognitive theory researchers, and clinicians, particularly those with an interest in clinical applications in trauma therapies.

---

## [Referee Report · Reviewer #1 (Public review)]

The authors attempted to replicate previous work showing that counterconditioning leads to more persistent reduction of threat responses, relative to extinction. They also aimed to examine the neural mechanisms underlying counterconditioning and extinction. They achieved both of these aims, and were able to provide some additional information, such as how counterconditioning impacts memory consolidation. Having a better understanding of which neural networks are engaged during counterconditioning may provide novel pharmacological targets to aid in therapies for traumatic memories. It will be interesting to follow up by examining the impact of varying amounts of time between acquisition and counterconditioning phases, to enhance replicability to real world therapeutic settings.

Major strengths

• This paper is very well written and attempts to comprehensively assess multiple aspects counterconditioning and extinction processes. For instance, the addition of memory retrieval tests is not core to the primary hypotheses, but provides additional mechanistic information on how episodic memory is impacted by counterconditioning. This methodical approach is commonly seen in animal literature, but less so in human studies.

• The Group x Cs-type x Phase repeated measure statistical tests with 'differentials' as outcome variables are quite complex, however the authors have generally done a good job of teasing out significant F test findings with post hoc tests and presenting the data well visually. It is reassuring that there is convergence between self-report data on arousal and valence and the pupil dilation response. Skin conductance is a notoriously challenging modality, so it is not too concerning that this was placed in the supplementary materials. Neural responses also occurred in logical regions with regards to reward learning.

• Strong methodology with regards to neuroimaging analysis, and physiological measures.

• The authors are very clear on documenting where there were discrepancies from their pre-registration and providing valid rationales for why.

Major Weaknesses

• The statistics showing that counterconditioning prevents differential spontaneous recovery are the weakest p values of the paper (and using one tailed tests, although this is valid due to directions being pre-hypothesised). This may be due to relatively small number of participants and some variability in responses.

---

## [Referee Report · Reviewer #2 (Public review)]

Summary:

The present study sets out to examine the impact of counterconditioning (CC) and extinction on conditioned threat responses in humans, particularly looking at neural mechanisms involved in threat memory suppression. By combining behavioral, physiological, and neuroimaging (fMRI) data, the authors aim to provide a clear picture of how CC might engage unique neural circuits and coding dynamics, potentially offering a more robust reduction in threat responses compared to traditional extinction.

Strengths:

One major strength of this work lies in its thoughtful and unique design - integrating subjective, physiological, and neuroimaging measures to capture the variouse aspects of counterconditiong (CC) in humans. Additionally, the study is centered on a well-motivated hypothesis and the findings have potentials for improving the current understanding of pathways associated with emotional and cognitive control.

The data presentation is systematic, and the results on behavioral and physiological measures fit well with the hypothesized outcomes. The neuroimaging results also provide strong support for distinct neural mechanisms underlying CC versus extinction.

Weaknesses:

Overall, this study is a well-conducted and thought-provoking investigation into counterconditioning, with strong potential to advance our understanding of threat modulation mechanisms. Two minor weaknesses concern the scope and decisions regarding analysis choices. First, while the findings are solid, the topic of counterconditioning is relatively niche and may have limited appeal to a broader audience. Expanding the discussion to connect counterconditioning more explicitly to widely studied frameworks in emotional regulation or cognitive control would enhance the paper's accessibility and relevance to a wider range of readers. This broader framing could also underscore the generalizability and broader significance of the results. In addition, detailed steps in the statistical procedures and analysis parameters seem to be missing. This makes it challenging for readers to interpret the results in light of potential limitations given the data modality and/or analysis choices.

Comments on revisions: My previous concerns and questions have been sufficiently addressed.

---

## [Referee Report · Reviewer #3 (Public review)]

In this manuscript, Wirz et al use neuroimaging (fMRI) to show that counterconditioning produces a longer lasting reduction in fear conditioning relative to extinction and appears to rely on the nucleus accumbens rather than the ventromedial prefrontal cortex. These important findings are supported by convincing evidence and will be of interest to researchers across multiple subfields, including neuroscientists, cognitive theory researchers, and clinicians.

In large part, the authors achieved their aims of giving a qualitative assessment of the behavioural mechanisms of counterconditioning versus extinction, as well as investigating the brain mechanisms. The results support their conclusions and give interesting insights into the psychological and neurobiological mechanisms of the processes that underlie the unlearning, or counteracting, of threat conditioning.

Strengths:

* Clearly written with interesting psychological insights

* Excellent behavioural design, well-controlled and tests for a number of different psychological phenomena (e.g. extinction, recovery, reinstatement, etc).

* Very interesting results regarding the neural mechanisms of each process.

* Good acknowledgement of the limitations of the study.

Weaknesses:

* I am not sure that the memories tested were truly episodic

* Twice as many female participants than males

Comments on revisions: I have no remaining concerns

---

## [Author Response]

The following is the authors’ response to the original reviews

**Public reviews:**

**Reviewer #1:**
The authors attempted to replicate previous work showing that counterconditioning leads to more persistent reduction of threat responses, relative to extinction. They also aimed to examine the neural mechanisms underlying counterconditioning and extinction. They achieved both of these aims and were able to provide some additional information, such as how counterconditioning impacts memory consolidation. Having a better understanding of which neural networks are engaged during counterconditioning may provide novel pharmacological targets to aid in therapies for traumatic memories. It will be interesting to follow up by examining the impact of varying amounts of time between acquisition and counterconditioning phases, to enhance replicability to real-world therapeutic settings.Major strengths· This paper is very well written and attempts to comprehensively assess multiple aspects of counterconditioning and extinction processes. For instance, the addition of memory retrieval tests is not core to the primary hypotheses but provides additional mechanistic information on how episodic memory is impacted by counterconditioning. This methodical approach is commonly seen in animal literature, but less so in human studies.· The Group x Cs-type x Phase repeated measure statistical tests with 'differentials' as outcome variables are quite complex, however, the authors have generally done a good job of teasing out significant F test findings with post hoc tests and presenting the data well visually. It is reassuring that there is a convergence between self-report data on arousal and valence and the pupil dilation response. Skin conductance is a notoriously challenging modality, so it is not too concerning that this was placed in the supplementary materials. Neural responses also occurred in logical regions with regard to reward learning.· Strong methodology with regards to neuroimaging analysis, and physiological measures.·The authors are very clear on documenting where there were discrepancies from their pre-registration and providing valid rationales for why.

We thank reviewer 1 for the positive feedback and for pointing out the strengths of our work. We agree that future research should investigate varying times between acquisition and counterconditioning to assess its success in real-life applications.

Major Weaknesses(1) The statistics showing that counterconditioning prevents differential spontaneous recovery are the weakest p values of the paper (and using one-tailed tests, although this is valid due to directions being pre-hypothesized). This may be due to a relatively small number of participants and some variability in responses. It is difficult to see how many people were included in the final PDR and neuroimaging analyses, with exclusions not clearly documented. Based on Figure 3, there are relatively small numbers in the PDR analyses (n=14 and n=12 in counterconditioning and extinction, respectively). Of these, each group had 4 people with differential PDR results in the opposing direction to the group mean. This perhaps warrants mention as the reported effects may not hold in a subgroup of individuals, which could have clinical implications.

General exclusion criteria are described on page 17. We have added more detailed information on the reasons for exclusion (see page 17). All exclusions were in line with pre-registered criteria. For the analysis, the reviewer is referring to (PDR analysis that investigated whether CC can prevent the spontaneous recovery of differential conditioned threat responses), 18 participants were excluded from this analysis: 2 participants did not show evidence for successful threat acquisition as was already indicated on page 17, and 16 participants were excluded due to (partially) missing data. We now explicitly mention the exclusion of the additional 16 participants on page 7 and have updated Figure 3 to improve visibility of the individual data points. Therefore, for this analysis both experimental groups consisted of 15 participants (total N=30).

It is true that in both groups a few participants show the opposite pattern. Although this may also be due to measurement error, we agree that it is relevant to further investigate this in future studies with larger sample sizes. It will be crucial to identify who will respond to treatments based on the principles of standard extinction or counterconditioning. We have added this point in the discussion on page 14.

**Reviewer #2:**
Summary:The present study sets out to examine the impact of counterconditioning (CC) and extinction on conditioned threat responses in humans, particularly looking at neural mechanisms involved in threat memory suppression. By combining behavioral, physiological, and neuroimaging (fMRI) data, the authors aim to provide a clear picture of how CC might engage unique neural circuits and coding dynamics, potentially offering a more robust reduction in threat responses compared to traditional extinction.Strengths:One major strength of this work lies in its thoughtful and unique design - integrating subjective, physiological, and neuroimaging measures to capture the various aspects of counterconditioning (CC) in humans. Additionally, the study is centered on a well-motivated hypothesis and the findings have the potential to improve the current understanding of pathways associated with emotional and cognitive control. The data presentation is systematic, and the results on behavioral and physiological measures fit well with the hypothesized outcomes. The neuroimaging results also provide strong support for distinct neural mechanisms underlying CC versus extinction.

We thank reviewer 2 for the feedback and for valuing the thoughtfulness that went into designing the study.

Weaknesses:(1) Overall, this study is a well-conducted and thought-provoking investigation into counterconditioning, with strong potential to advance our understanding of threat modulation mechanisms. Two main weaknesses concern the scope and decisions regarding analysis choices. First, while the findings are solid, the topic of counterconditioning is relatively niche and may have limited appeal to a broader audience. Expanding the discussion to connect counterconditioning more explicitly to widely studied frameworks in emotional regulation or cognitive control would enhance the paper's accessibility and relevance to a wider range of readers. This broader framing could also underscore the generalizability and broader significance of the results. In addition, detailed steps in the statistical procedures and analysis parameters seem to be missing. This makes it challenging for readers to interpret the results in light of potential limitations given the data modality and/or analysis choices.

In this updated version of the manuscript, we included the notion that extinction has been interpreted as a form of implicit emotion regulation. In addition to our discussion on active coping (avoidance), we believe that our discussion has an important link to the more general framework of emotion regulation, while remaining within the scope of relevance. Please see pages 14 and 15 for the changes. In addition to being informative to theories of emotion regulation, our findings are also highly relevant for forms of psychotherapy that build on principles of counterconditioning (e.g. the use of positive reinforcement in cognitive behavioral therapy), as we point out in the introduction. We believe this relevance shows that counterconditioning is more than a niche topic. In line with the recommendation from reviewer 2, we added more details and explanations to the statistical procedures and analyses where needed (see responses to recommendations).

**Reviewer #3:**
Summary:In this manuscript, Wirz et al use neuroimaging (fMRI) to show that counterconditioning produces a longer lasting reduction in fear conditioning relative to extinction and appears to rely on the nucleus accumbens rather than the ventromedial prefrontal cortex. These important findings are supported by convincing evidence and will be of interest to researchers across multiple subfields, including neuroscientists, cognitive theory researchers, and clinicians.In large part, the authors achieved their aims of giving a qualitative assessment of the behavioural mechanisms of counterconditioning versus extinction, as well as investigating the brain mechanisms. The results support their conclusions and give interesting insights into the psychological and neurobiological mechanisms of the processes that underlie the unlearning, or counteracting, of threat conditioning.Strengths:· Mostly clearly written with interesting psychological insights· Excellent behavioural design, well-controlled and tests for a number of different psychological phenomena (e.g. extinction, recovery, reinstatement, etc).· Very interesting results regarding the neural mechanisms of each process.· Good acknowledgement of the limitations of the study.

We thank reviewer 3 for the detailed feedback and suggestions.

Weaknesses:(1) I think the acquisition data belongs in the main figure, so the reader can discern whether or not there are directional differences prior to CC and extinction training that could account for the differences observed. This is particularly important for the valence data which appears to differ at baseline (supplemental figure 2C).

Since our design is quite complex with a lot of results, we left the fear acquisition results as a successful manipulation check in the Supplementary Information to not overload the reader with information that is not the main focus of this manuscript. If the editor would like us to add the figure to the main text, we are happy to do so. During fear acquisition, both experimental groups showed comparable differential conditioned threat responses as measured by PDRs and SCRs. Subjective valence ratings indeed differed depending on CS category. Importantly, however, the groups only differed with respect to their rating to the CS- category, but not the CS+ category, which suggests that the strength of the acquired fear is similar between the groups. To make sure that these baseline differences cannot account for the differences in valence after CC/Ext, we ran an additional group comparison with differential valence ratings after fear acquisition added as a covariate. Results show that despite the baseline difference, the group difference in valence after CC/Ext is still significant (main effect Group: F_(1,43)_=7.364, p=0.010, η^2^=0.146). We have added this analysis to the manuscript (see page 7).

(2) I was confused in several sections about the chronology of what was done and when. For instance, it appears that individuals went through re-extinction, but this is just called extinction in places.

We understand that the complexity of the design may require a clearer description. We therefore made some changes throughout the manuscript to improve understanding. Figure 1 is very helpful in understanding the design and we therefore refer to that figure more regularly (see pages 6-7). We also added the time between tasks where appropriate (e.g. see page 7). Re-extinction after reinstatement was indeed mentioned once in the manuscript. Given that the reinstatement procedure was not successful (see page 9), we could not investigate re-extinction and it is therefore indeed not relevant to explicitly mention and may cause confusion. We therefore removed it (see page 12).

(3) I was also confused about the data in Figure 3. It appears that the CC group maintained differential pupil dilation during CC, whereas extinction participants didn't, and the authors suggest that this is indicative of the anticipation of reward. Do reward-associated cues typically cause pupil dilation? Is this a general arousal response? If so, does this mean that the CSs become equally arousing over time for the CC group whereas the opposite occurs for the extinction group (i.e. Figure 3, bottom graphs)? It is then further confusing as to why the CC group lose differential responding on the spontaneous recovery test. I'm not sure this was adequately addressed.

Indeed, reward and reward anticipation also evoke an increase in pupil dilation. This was an important reason for including a separate valence-specific response characterization task. Independently from the conditioning task, this task revealed that both threat and reward-anticipation induced strong arousal-related PDRs and SCRs. This was also reflected in the explicit arousal ratings, which were stronger for both the shock-reinforced (negative valence) and reward-reinforced (positive valence) stimuli. Therefore, it is not surprising that reward anticipation leads to stronger PDRs for CS+ (which predict reward) compared to CS- stimuli (which do not predict reward) during CC, but is reduced during extinction due to a decrease in shock anticipation. During the spontaneous recovery test, a return of stronger PDRs for CS+ compared to CS- stimuli in the standard extinction group can only reflect a return of shock anticipation. Importantly, the CC group received no rewards during the spontaneous recovery task and was aware of this, so it is to be expected that the effect is weakened in the CC group. However, CS+ and CS- items were still rated of similar valence and PDRs did not differ between CS+ and CS- items in the CC group, whereas the Ext group rated the CS+ significantly more negative and threat responses to the CS+ did return. It therefore is reasonable to conclude that associating the CS+ with reward helps to prevent a return of threat responses. We have added some clarifications and conclusions to this section on page 8.

(4) I am not sure that the memories tested were truly episodic

In line with previous publications from Dunsmoor et al.[1-4], our task allows for the investigation of memory for elements of a specific episode. In the example of our task, retrieval of a picture probes retrieval of the specific episode, in which the picture was presented. In contrast, fear retrieval relies on the retrieval of the category-threat association, which does not rely on retrieval of these specific episodic elements, but could be semantic in nature, as retrieval takes place at a conceptual level. We have added a small note on what we mean with episodic in this context on page 4. We do agree that we cannot investigate other aspects of episodic memories here, such as context, as this was not manipulated in this experiment.

(5) Twice as many female participants than males

It is indeed unfortunate that there is no equal distribution between female and male participants. Investigating sex differences was not the goal of this study, but we do hope that future studies with the appropriate sample sizes are able to investigate this specifically. We have added this to the limitations of this study on page 17.

(6) No explanation as to why shocks were varied in intensity and how (pseudo-randomly?)

The shock determination procedure is explained on pages 18-19 (Peripheral stimulation). As is common in fear conditioning studies in humans (see references), an ascending staircase procedure was used. The goal of this procedure is to try and equalize the subjective experience of the electrical shocks to be “maximally uncomfortable but not painful”.

**Recommendations for the authors:**

**Reviewer #1:**
Very well written. No additional comments

We thank reviewer 1 for valuing our original manuscript version. To further improve the manuscript, we adapted the current version based on the reviewer’s public review (see response to reviewer #1 public review comment 1).

**Reviewer #2:**
(1) I feel that more justification/explanation is needed on why other regions highly relevant to different aspects of counterconditioning (e.g., threat, memory, reward processing) were not included in the analyses.

We first performed whole-brain analyses to get a general idea of the different neural mechanisms of CC compared to Ext. Clusters revealing significant group differences were then further investigated by means of preregistered ROI analyses. We included regions that have previously been shown to be most relevant for affective processing/threat responding (amygdala), memory (hippocampus), reward processing (NAcc) and regular extinction (vmPFC). We restricted our analyses to these most relevant ROIs as preregistered to prevent inflated or false-positive findings[5]. Beyond these preregistered ROIs, we applied appropriate whole-brain FEW corrections. The activated regions are listed in Supplementary Table 1 and include additional regions that were expected, such as the ACC and insula.

(2) Were there observed differences across participants in the experiment? Any information on variance in the data such as how individual differences might influence these findings would provide a richer understanding of counterconditioning and increase the depth of interpretation for a broad readership.

We agree that investigating individual differences is crucial to gain a better understanding of treatment efficacy in the framework of personalized medicine. Specifically, future research should aim to identify factors that help predict which treatment will be most effective for a particular patient. The results of this study provide a good basis for this, as we could show that the vmPFC in contrast to regular extinction, is not required in CC to improve the retention of safety memory. Therefore, this provides a viable option for patients who are not responding to treatments that rely on the vmPFC. In addition, as noted by Reviewer 1, in both groups a few participants show the opposite pattern (see Figure 3). It will be crucial to identify who will respond to treatments based on the principles of standard extinction or counterconditioning. We have added this point in the discussion on page 14.

(3) While most figures are informative and clear, Figure 3 would benefit from detailed axis labels and a more descriptive caption. Currently, it is challenging to navigate the results presented to support the findings related to differential PDRs. A supplementary figure consolidating key patterns across conditions might also further facilitate understanding of this rather complicated result.

We have made some changes to the figure to improve readability and understanding. Specifically, we changed the figure caption to “Change from last 2 trials CC/Ext to first 2 trials Spontaneous recovery test”, to give more details on what exactly is shown here. We also simplified the x-axis labels to “counterconditioning”, “recovery test” and “extinction”. With the addition of a clearer figure description, we hope to have improved understanding and do not think that another supplemental figure is needed.

(4) Additional details on the statistical tests are needed. For example, please clarify whether p-values reported were corrected across all experimental conditions. Also, it would be helpful for the authors to discuss why for example repeated measures ANOVA or mixed-effects conditions were not used in this study. Might those tests not capture variance across participants' PDRs and SCRs over time better?

We added that significant interactions were followed by Bonferroni-adjusted post-hoc tests where applicable (see page 21). We have used repeated measures ANOVAs to capture early versus late phases of acquisition and CC/extinction, as well as to compare late CC/extinction (last 2 trials) compared to early spontaneous recovery (first 2 trials) as is often done in the literature. A trial-level factor in a small sample would cost too many degrees of freedom and is not expected to provide more information. We have added this information and our reasoning to the methods section on page 21.

**Reviewer #3:**
(1) Suggest putting acquisition data into the main figures. In fact many of the supplemental figures could be integrated into the main figures in my opinion.

See response to reviewer #3 public review comment 1.

(2) Include explanations for why shock intensity was varied

See response to reviewer #3 public review comment 6.

(3) Include a better explanation for the change in differential responding from training to spontaneous recovery in the CC group (I think the loss of such responding in extinction makes more sense and is supported by the notion of spontaneous recovery, but I'm not sure about the loss in the CC group. There is some evidence from the rodent literature - which I am most familiar with - regarding a loss in contextual gradient across time which could account for some loss in specificity, could it be something like this?).

See response to reviewer #3 public review comment 3.

If we understand the reviewer correctly in that the we see a loss of differential responding due to a generalization to the CS-, this would imply an increase in responding to the CS-, which is not what we see. Our data should therefore be correctly interpreted as a loss of the specific response to the CS+ from the CC phase to the recovery test. Therefore, there is no spontaneous recovery in the CC group, and also not a non-specific recovery. To clarify this we relabeled Figure 3 by indicating “recovery test” instead of “spontaneous recovery”.

(4) Is there a possibility that baseline differences, particularly that in Supplemental Figure 2C, could account for later differences? If differences persist after some transformation (e.g. percentage of baseline responding) this would be convincing to suggest that it doesn't.

See response to reviewer #3 public review comment 1.

(5) As I mentioned, I got confused by the chronology as I read through. Maybe mention early on when reporting the spontaneous recovery results that testing occurred the next day and that participants were undergoing re-extinction when talking about it for the second time.

See response to reviewer #3 public review comment 2.

(6) Page 8 - I was confused as to why it is surprising that the CC group were more aroused than the extinction group, the latter have not had CSs paired with anything with any valence, so doesn't this make sense? Or perhaps I am misunderstanding the results - here in text the authors refer back to Figure 2B, but I'm not sure if this is showing data from the spontaneous recovery test or from CC/extinction. If it is the latter, as the caption suggests, why are the authors referring to it here?

Participants in the CC group showed increased differential self-reported arousal after CC, whereas arousal ratings did not differ between CS+ and CS- items after extinction. We interpret this in line with the valence and PDR results as an indication of reward-induced arousal. At the start of the next day, however, participants from the CC and extinction groups gave comparable ratings. It may therefore be surprising why participants in the CC group do not still show stronger ratings since nothing happened between these two ratings besides a night’s sleep (see design overview in Figure 1A). We removed the “suprisingly” to prevent any confusion.

(7) I suggest that the authors comment on whether there were any gender differences in their results.

See response to reviewer #3 public review comment 5.

(8) The study makes several claims about episodic memory, but how can the authors be sure that the memories they are tapping into are episodic? Episodic has a very specific meaning - a biographical, contextually-based memory, whereas the information being encoded here could be semantic. Perhaps a bit of clarification around this issue could be helpful.

See response to reviewer #3 public review comment 4.

References

(1) Dunsmoor, J. E. & Kroes, M. C. W. Episodic memory and Pavlovian conditioning: ships passing in the night. Curr Opin Behav Sci 26, 32-39 (2019). https://doi.org/10.1016/j.cobeha.2018.09.019

(2) Dunsmoor, J. E. et al. Event segmentation protects emotional memories from competing experiences encoded close in time. Nature Human Behaviour 2, 291-299 (2018). https://doi.org/10.1038/s41562-018-0317-4

(3) Dunsmoor, J. E., Murty, V. P., Clewett, D., Phelps, E. A. & Davachi, L. Tag and capture: how salient experiences target and rescue nearby events in memory. Trends Cogn Sci 26, 782-795 (2022). https://doi.org/10.1016/j.tics.2022.06.009

(4) Dunsmoor, J. E., Murty, V. P., Davachi, L. & Phelps, E. A. Emotional learning selectively and retroactively strengthens memories for related events. Nature 520, 345-348 (2015). https://doi.org/10.1038/nature14106

(5) Gentili, C., Cecchetti, L., Handjaras, G., Lettieri, G. & Cristea, I. A. The case for preregistering all region of interest (ROI) analyses in neuroimaging research. Eur J Neurosci 53, 357-361 (2021). https://doi.org/10.1111/ejn.14954